# LIKELIHOOD-FREE INFERENCE OF PHYLOGENETIC TREE POSTERIOR DISTRIBUTIONS

## ABSTRACT

Phylogenetic inference, the task of reconstructing how related sequences evolved from common ancestors, is a central objective in evolutionary genomics. The current state-of-the-art methods exploit probabilistic models of sequence evolution along phylogenetic trees, by searching for the tree maximizing the likelihood of observed sequences, or by estimating the posterior of the tree given the sequences in a Bayesian framework. Both approaches typically require to compute likelihoods, which is only feasible under simplifying assumptions such as independence of the evolution at the different positions of the sequence, and even then remains a costly operation. Here we present the first likelihood-free inference method for posterior distributions over phylogenies. It exploits a novel expressive encoding for pairs of sequences, and a parameterized probability distribution factorized over a succession of subtree merges. The resulting network provides accurate estimates of the posterior distribution outperforming both state-of-the-art maximum likelihood methods and a previous likelihood-free method for point estimation. It opens the way to fast and accurate phylogenetic inference under models of sequence evolution beyond those amenable to current likelihood-based inference methods.

## 1 INTRODUCTION

The genomes of living species evolve over time through a process that involves mutations and selection. Reconstructing the evolutionary history of a set of contemporaneous sequences is a central task in genomics (Kapli et al., 2020): it is used to understand how extant species have evolved from common ancestors (Álvarez Carretero et al., 2022), how bacterial resistances to drugs have emerged and been disseminated (Aminov & Mackie, 2007), and how epidemics are spreading Hadfield et al. (2018). A key object in this endeavor is the phylogeny, a bifurcating tree summarizing the succession of transformations of the sequence that lead to the current observed diversity from a single ancestor.

Modern phylogenetic reconstruction is now dominated by approaches based on probabilistic models of sequence evolution. Model-based approaches offer a principled framework for inference by making assumptions about evolutionary processes transparent, while also providing methods for model checking, criticism and iterative model elaboration (Gelman et al., 2004). Models of sequence evolution are typically continuous time Markov processes parameterized by the phylogeny and by the rates at which amino-acids or nucleotides undergo substitutions. Existing reconstruction methods look for the phylogeny maximizing the likelihood of the sequences (Minh et al., 2020; Price et al., 2010) or, in a Bayesian perspective, aim at sampling from the posterior distribution of the phylogeny given the sequences through Monte Carlo strategies (Huelsenbeck & Ronquist, 2001; Höhna et al., 2016; Bouchard-Côté et al., 2012) or approximate this posterior distribution through variational inference (Zhang & IV, 2019; Koptagel et al., 2022; Zhou et al., 2024; Duan et al., 2024).

Across all these approaches, a major hurdle is the computational cost of evaluating the likelihood function, which is required either for maximizing the likelihood, computing acceptance probabilities in sampling strategies, or computing the evidence lower bound (ELBO) objective in the case of variational inference. For any single phylogeny, the likelihood is computed through a costly pruning algorithm (Felsenstein, 1981) and many such evaluations are required to heuristically explore the set of possible tree topologies—$(2N-5)!!$ for a phylogeny over $N$ leaves. Furthermore, just making this computation feasible has resulted in a focus on probabilistic models that make simplifying assumptions such as independence and identical distribution of the evolutionary process at each position

in the sequence, or the absence of natural selection. These simplifications are known to produce unrealistic sets of sequences (Trost et al., 2024), artifacts in reconstructed phylogenies (Telford et al., 2005), and impede our ability to understand the evolutionary history of living species.

Simulation-based or likelihood-free inference has emerged as a powerful paradigm for estimation under probabilistic models under which likelihood evaluations are intractable but sampling is cheap (Cranmer et al., 2020; Lueckmann et al., 2021). This paradigm has leveraged advances in deep learning to approximate posterior distributions by neural networks trained over data simulated under probabilistic models (Greenberg et al., 2019; Lueckmann et al., 2021). Among these methods, neural posterior estimation (NPE, Lueckmann et al., 2021) defines a family of distributions parameterized by a neural network whose weights are then optimized to approximate the posterior. In addition to working around the need for likelihood evaluation, NPE is amortized: training the network can take time but performing inferences with the trained network is typically very fast. By contrast, existing variational inference and Monte Carlo samplers require new computation everytime a new inference is done. On the other hand, NPE require to define a parameterized family of distributions that is appropriate for the posteriors of phylogenies given sequences, which is not straightforward.

Here we introduce Phyloformer 2, an NPE for phylogenetic reconstruction, with the following contributions:

- We propose a parameterized family of posterior distributions on phylogenies given a set of sequences, factorized through a succession of pairwise mergings. Optimizing the weights of our network to maximize the log-probability within this family yields a likelihood-free estimate of the corresponding posterior that can then be used for sampling trees. To our knowledge, this is the first likelihood-free posterior estimation method trained end-to-end from sequences to the phylogeny beyond quartets.

- To extract the parameters of the approximate posterior distribution from the input sequences, we introduce a novel evoPhyloFormer (evoPF) architecture akin to the EvoFormer module used in Alphafold 2 (Jumper et al., 2021), that is both more scalable and expressive than the one used in Phyloformer. The overall architecture allows us to scale up to over 200 sequences of length 500 or more than 300 sequences of length 250 on a single V100 GPU with 16Gb of VRAM.

- On data generated under a probabilistic model of sequence evolution with tractable likelihood, Phyloformer 2 outperforms both state-of-the-art likelihood-based and likelihood-free reconstruction methods in topological accuracy and produces estimates of the posterior compared to MCMC samples. Because it is likelihood-free, it can also be trained to produce estimates under models with intractable likelihoods, in which case the performance gap with—misspecified—likelihood-based estimators further increases.

- Because Phyloformer 2 is amortized, once trained, it can perform inference 1 to 2 orders of magnitude faster than the—less accurate—state-of-the-art likelihood-based estimators.

## RELATED WORK

Initial attempts to phylogenetic NPE have restricted themselves to quartets, *i.e.*, topologies over four leaves, allowing them to cast the problem as a classification over the three possible topologies (Suvorov et al., 2019; Zou et al., 2020; Tang et al., 2024). In this case, a vector embedding of the input sequences was extracted by a neural network and used to produce three scalar outputs, and the probability of each topology was simply modeled as a softmax over these three outputs. Because the number of possible topologies grows super-exponentially with the number of leaves, this strategy cannot be generalized to larger numbers of sequences and even if it could, treating all topologies as separate classes would disregard the fact that some are more similar than others. In addition, Grosshauser M (2021) re-evaluated the method of Zou et al. (2020) and showed that it underperformed on more difficult tasks with short sequences and long evolution times.

Alternatively, Nesterenko et al. (2025) proposed Phyloformer, a likelihood-free inference method defined for any number of leaves. Phyloformer doesn't estimate phylogenies, but evolutionary distances, *i.e.*, sum of branch lengths on the phylogeny between pairs of leaves. Phylogenies can be reconstructed from such distances, for example with the neighbor joining algorithm (NJ, Saitou & Nei, 1987) , but the authors observed that this strategy led to a limited topological reconstruction

accuracy. In addition, Phyloformer only allows for point estimates—namely the median of the posterior distribution of evolutionary distances rather than the entire distribution. Finally, the network applied axial self-attention (Ho et al., 2019) to all pairs of sequences, which led to a large memory footprint even using a linear approximation of self-attention (Katharopoulos et al., 2020). The authors reported that using axial self-attention on sequences instead of pairs as in the MSA transformer (Rao et al., 2021) improved the scalability but dramatically decreased the reconstruction accuracy.

Phyloformer 2 addresses each of these limitations of Phyloformer by estimating the entire posterior distribution of phylogenies—instead of a single estimate of evolutionary distances— thanks to its BayesNJ module, and by using a novel evoPF encoding that is more expressive than the MSA transformer but more lightweight than Phyloformer.

## 2 BACKGROUND AND NOTATION

### 2.1 NOTATION

Let $x = \{x_1, \ldots, x_N\}$ be a set of $N$ sequences of $L$ letters in some fixed alphabet representing organic compounds (*e.g.*, 20 possible amino acids for proteins, 4 possible nucleotides for DNA). We assume that the sequences are aligned, *i.e.*, correspond to $N$ initial sequences of possibly different lengths whose positions were matched to minimize some score quantifying how similar all sequences are at each position, potentially by introducing gaps denoted by a special character in the alphabet (Kapli et al., 2020).

A phylogeny $\theta = (\tau, \ell)$ over $x$ is an unrooted binary tree $\tau$ with $N$ leaves and a set $\ell$ of branch lengths in $\mathbb{R}_+^{2N-3}$. The tree $\tau$ can equivalently be represented by a succession of merges. Intuitively, given $N$ species, one chooses two species, pair them to form a cherry, replace them by a single species representing their ancestor, and proceed recursively with the $N-1$ resulting species until the entire tree has been produced. Formally, we denote this succession of merges as $\left\{m^{(k)}\right\}_{k=1}^{N-3}$ where $m^{(k)} \in \mathcal{C}^{(k)} \triangleq \left\{\left(v_i^{(k)}, v_j^{(k)}\right) \in \mathcal{S}_{(k)}^2 | i \neq j\right\}$ is a pair of two distincts elements in $\mathcal{S}_{(k)}$ the set of "mergeable" nodes at the $k^{th}$ merge, *i.e.*, either leaves or internal nodes whose children were already merged, and $\mathcal{C}^k$ is the set of candidate merges at step $k$. In order to build $\mathcal{S}_{(k)}$, we define $\mathcal{S}_{(1)}$ to be the set of leaves in $\tau$ and for $k > 1$, $\mathcal{S}_{(k)} \triangleq \left\{\left\{\mathcal{S}_{(k-1)} \cup u^{(k-1)}\right\} \setminus \left(v_i^{(k-1)}, v_j^{(k-1)}\right)\right\}$, and $u^{(k)}$ is the common neighbor in $\tau$ to the nodes merged in $m^{(k)}$—which is always defined because we start from leaves and recursively replace pairs of elements by their common neighbor in the binary tree.

We further denote $\ell^{(k)} = \left\{\ell_i^{(k)}, \ell_j^{(k)}\right\} \in \mathbb{R}_+^2$ the $k$-th cherry, *i.e.*, the set of two branch lengths connecting $m^{(k)}$ to $u^{(k)}$. In our context, the $N$ leaves will represent the taxa with sequences in $x$, $u^{(k)}$ is the common ancestor of $m^{(k)}$, and $\ell^{(k)}$ is the set of evolutionary times between this ancestor and its two descendants.

### 2.2 NEURAL POSTERIOR ESTIMATION

For a probabilistic model $p(x|\theta)$ of the data $x$ given the parameter $\theta$ and a prior distribution $p(\theta)$, NPE provides a way to estimate the posterior $p(\theta|x)$ in cases where evaluating $p(x|\theta)$ for a given $(x, \theta)$ is too costly or intractable, but where it is possible to sample from this model. It relies on a family of distributions $q_\psi(\theta|x)$ whose parameters $\psi$ are provided by a neural network acting on the data $x$—by abuse of notation we denote $\psi(x)$ the parameters output by this neural network for an input $x$. NPE builds its approximation of $p(\theta|x)$ by looking for the $q_\psi$ minimizing the average Kullback-Leibler (KL) divergence with the true posterior:

$$\mathbb{E}_{p(x)}\left[\mathrm{KL}(q_\psi(\theta|x)||p(\theta|x))\right] = \mathbb{E}_{p(x)}\left[\mathbb{E}_{p(\theta|x)}\left[\log p(\theta|x) - \log q_\psi(\theta|x))\right]\right],$$

where the average is taken over the marginal $p(x) = \int p(x|\theta)p(\theta)d\theta$. Since the $p(\theta|x)$ term does not depend on $\psi$, minizing this average KL divergence with respect to $\psi$ is equivalent to maximizing the average approximate log-likelihood $q_\psi$ over the joint distribution $p(x, \theta) = p(x|\theta)p(\theta)$ (see *e.g.*, Radev et al., 2020). This is generally achieved by minimizing a Monte Carlo approximation $\sum_{i=1}^n \log q_{\psi(x_i)}(\theta_i|x_i)$ of this average, built over a large number of examples $\{(x_i, \theta_i)\}_{i=1}^n$ sampled from $p(x, \theta)$ by successively sampling a $\theta_i$ from the prior and an $x_i$ from the model given $\theta_i$.

Consequently, NPE is guaranteed to converge to the true posterior $p(\theta|x)$, provided that the family $q_\psi$ is expressive enough to represent $p(\theta|x)$ and that the optimization algorithm finds the optimum. More precisely, its approximation error depends both on the expressivity of the chosen family of distributions, *i.e.*, on the average KL divergence $\mathbb{E}_{p(x)}\left[\mathrm{KL}(q_{\psi^*(x)}(\theta|x)||p(\theta|x))\right]$ between the best distribution $q_{\psi^*(x)}$ in the family, and the true posterior for each $x$, and on the expressivity of the neural network, *i.e.*, on its ability to map every $x$ to the corresponding best parameters $\psi^*(x)$.

## 3 METHODS

Phyloformer 2 combines two novel modules. The first one, evoPF, encodes distribution parameters $\psi(x)$ from a set of aligned related sequences $x$, which is sometimes refered to as a multiple sequence alignment (MSA). The second one, BayesNJ, defines a family of posterior distributions $q_{\psi(x)}\left(\theta = (\tau, \ell)|x\right)$ on phylogenetic trees, parameterized by the output of evoPF. Figure 1a-b represents the overall architecture.

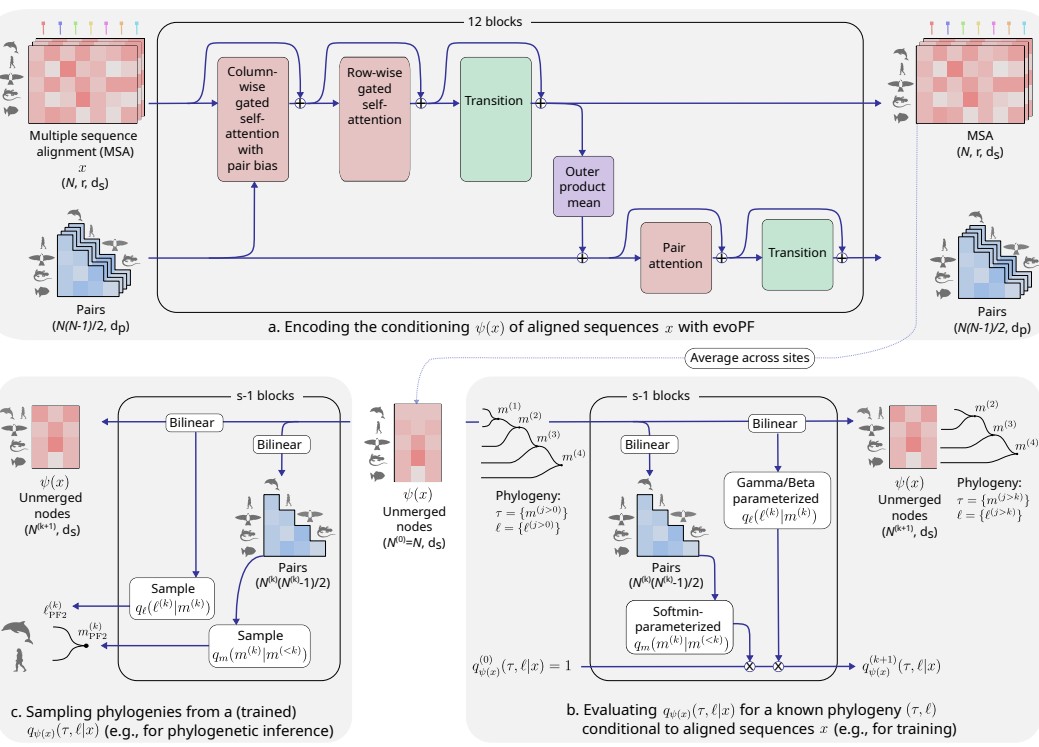

Figure 1: Architecture of Phyloformer 2. ***Panel a***: evoPF, an EvoFormer-inspired module updating $N \times L$ embeddings for a set of aligned sequences (MSA) $x$ and $N(N-1)/2$ embeddings for pairs of sequences. Each of its blocks applies self-attention within both the MSA and representation, and ensures information sharing between them. After 12 blocks, we extract one embedding for each sequence by averaging the MSA embeddings across sites. ***Panel b***: BayesNJ (Algorithm S.10) computes the posterior probability of a phylogeny given an MSA represented by the sequence and pair embeddings provided by evoPF. The probability is a product over a recursive operation where two taxa are merged into their parent, and the taxon representation is updated accordingly. ***Panel c***: at inference time, we apply the same succession of operations as for evaluating the probability, but either sampling or taking the modes of the distributions (Algorithm S.11).

### 3.1 ENCODING TREE DISTRIBUTION PARAMETERS WITH EVOPF

Our encoder should process a set $x$ of aligned sequences and be expressive enough to capture sufficient information on their evolutionary relationships. Nesterenko et al. (2025) encoded $x$ with

one embedding for each aligned position in each pair of aligned sequences. This approach avoided the need to flatten information across positions, while maintaining a representation at the pair level, consistently with their predicting pairwise evolutionary distances. They noted that tuning down the architecture to one embedding for each position in each sequence—boiling down to the MSA transformer (Rao et al., 2021)—dramatically improved the method scalability but strongly affected the accuracy of their trained network. Inspired by this trade-off, we introduce evoPF, an encoder that maintains a single embedding per pair, and a separate representation with one embedding per position within each sequence (Figure 1a).

EvoPF is a transpose version of the EvoFormer module in Alphafold 2 Jumper et al. (2021)—whose objective was to capture spatial distances between pairs of aligned positions rather than evolutionary distances between pairs of homologous sequences—with a few simplifications. The MSA stack maintains an embedding for each position within each sequence in $x$. It relies on axial attention, alternating one layer of column-wise and one layer of row-wise gated self-attention. The column-wise mechanism takes each sequence separately, and applies self-attention between embeddings of all its positions, allowing the information to flow within each sequence. Symmetrically, the row-wise mechanism lets the information flow between sequences by treating each position separately and applying self-attention over the embeddings of all sequences at this position. The two self-attention layers are followed by a transition layer applying a linear function and a ReLU to each embedding in the alignment $x$ separately.

Parallel to this MSA stack, we maintain an embedding for each pair of sequences in $x$, updated in each evoPF block by applying self-attention between the embeddings of all pairs, followed by a transition layer like in the MSA stack. This diverges from EvoFormer which relied on triangular attention where pair $(i, j)$ only attended to pairs $(i, k)$ and $(k, j)$. The MSA stack affects the pair stack through the addition of an outer product mean of sequence embeddings to the pair embeddings. Conversely, the pair stack affects the MSA stack by biasing its column-wise attention—*i.e.*, we compute a bias term for each pair of sequences by applying a linear function to each pair embedding, and add this bias to the attention scores in the column-wise attention layer, before we apply the usual softmax. We provide a complete description of evoPF in Algorithms S.1 to S.9 in Appendix A.1.

### 3.2 Defining a proper probability distribution over phylogenies with BayesNJ

After 12 evoPF blocks, we average the MSA embeddings across positions, yielding a single vector embedding per sequence. These embeddings constitute our encoding $\psi(x)$ of the sequences $x$, and our next goal is to define a family of distributions of phylogenies conditional on $x$, whose parameters are functions of $\psi(x)$. To this end, we define $q_{\psi(x)}(\theta = (\tau, \ell)|x)$ factorized over the succession of merges in $\tau$, *i.e.* $q_{\psi(x)}(\theta = (\tau, \ell)|x) = \Pi_{k=1}^{2N-3} q_m(m^{(k)}|m^{(<k)}) q_\ell(\ell^{(k)}|m^{(k)}, m^{(<k)})$, where $m^{(<k)}$ denotes the set of merges with indices smaller than $k$, and where the probability of each merge is further decomposed into a topological factor $q_m$ representing the probability of merging these two particular clades and a branch-length factor $q_\ell$ factor representing the probability of observing these evolutionary distances between the merged clades and their parents. Both $q_m$ and $q_\ell$ are parameterized by $\psi(x)$ as detailed in 3.3. A caveat of this factorization across $2N - 3$ merges is that most phylogenies $\theta$ can be obtained by several distinct successions of merges from the leaves. For example, a balanced binary tree with four leaves $a, b, c, d$ merging $(a, b)$ and $(c, d)$ can be obtained by merging any of the two groups first, and these two orders have no reason to lead to the same $\Pi_{k=1}^{2N-3} q_m(m^{(k)}|m^{(<k)}) q_\ell(\ell^{(k)}|m^{(k)}, m^{(<k)})$ in general. Properly defining a probability distribution over phylogenies therefore requires to sum over all possible orders of merge, which is not feasible even for moderately large $N$.

Alternatively, we must ensure that for a given phylogeny our distribution assigns a non-zero probability to a single merge order and, to be able to evaluate the probability of any phylogeny, that this order can be recovered efficiently from the phylogeny. We achieve this by designing a canonical merge order such that (i) we can guarantee that our sampling procedure always generates merges in this order and (ii) our evaluation procedure always processes merges in this order. Point (i) requires that the merge order does not depend on the stochastic parts of the sampling procedure, point (ii) requires that we can recover the canonical order efficiently for any given phylogeny. We achieve these two points by ensuring that at every step $k$, we select the merge corresponding to the two closest nodes currently available in $\mathcal{S}_{(k)}$—*i.e.*, whose children have already been merged. Conversely when

sampling a phylogeny $(\tau, \ell)$ from our $q_{\psi(x)}(\tau, \ell|x)$, we ensure that the distance between merged nodes $(i, j)$ is larger than the distance between any previous merges done while $(i, j)$ was a possible merge—*i.e.*, after their children were merged. A similar strategy was used in the likelihood-based sequential Monte Carlo sampler Bouchard-Côté et al. (2012).

### 3.3 BAYESNJ PARAMETERIZATION OF TOPOLOGICAL AND BRANCH-LENGTH COMPONENTS

We parameterize the topological component of $q_{\psi}$ by a softmin across pairwise scores computed from a symmetric bilinear function over embeddings of pairs of mergeable nodes at step $k$:

$$q_m\left(m^{(k)}|m^{(<k)}\right) = \frac{e^{-\text{score}_{m^{(k)}}}}{\sum_{m' \in \mathcal{C}^{(k)}} e^{-\text{score}_{m'}}}, \text{ and } \forall m = (u, v) \in \mathcal{C}^{(k)}, \text{score}_m = \mathbf{v}_u^\top A \mathbf{v}_v, \quad (1)$$

where the symmetric matrix $A$ is a learnable parameter and $\mathbf{v}_u$ denotes the embedding for node $u$ in the encoding $\psi(x)$. At $k = 1$ these embeddings are those provided by evoPF, and at each successive merge we remove two current leaves and create an embedding for their ancestor which becomes a new leaf (Figure 1b, Algorithm S.10). The topological component $q_m(m^{(k)}|m^{(<k)})$ is therefore a probabilistic version of the $\arg\min$ over pairwise distances that is used in classical hierarchical reconstruction algorithms such as NJ. Furthermore, $q_m$ relies on learnable scores rather than estimates of the evolutionary distances.

In order to ensure that each sampled merge $m^{(k)}$ leads to a longer cherry (sum of branch lengths) than those in all merges previously selected while $m^{(k)}$ was a possible choice—as decided in 3.2 to ensure that $p_{\psi(x)}$ is a proper probability distribution over phylogenies—we re-parameterize the two branch lengths $\left(\ell_i^{(k)}, \ell_j^{(k)}\right) \in \ell^{(k)}$ as their sum $s^{(k)} = \ell_i^{(k)} + \ell_j^{(k)}$ and ratio $r^{(k)} = \ell_i^{(k)}/s^{(k)}$. We update a matrix of constraints as we merge pairs indicating the minimal length that the sum of branch lengths at any new merge must attain for the sampled tree to be consistent with our canonical order. We then model the probability $q_s\left(s^{(k)}|m^{(k)}, m^{(<k)}\right)$ as a Gamma distribution shifted by the constraint, and the probability $q_r\left(r^{(k)}|m^{(k)}, m^{(<k)}\right)$ of the branch length ratio as a Beta distribution, *i.e.*,

$$q_s\left(s^{(k)}|m^{(\leq k)}\right) = c_{m^{(k)}} + \text{Gamma}\left(\alpha_\text{G}^{(k)}, \lambda_\text{G}^{(k)}\right), \; q_r\left(r^{(k)}|m^{(\leq k)}\right) = \text{Beta}\left(\alpha_\text{B}^{(k)}, \beta_\text{B}^{(k)}\right), \quad (2)$$

where $c_{m^{(k)}}$ is the current constraint on the sum of branch lengths for merge $m^{(k)}$ and whose parameters $\left(\alpha_\text{G}^{(k)}, \lambda_\text{G}^{(k)}, \alpha_\text{B}^{(k)}, \beta_\text{B}^{(k)}\right)$ are produced by bilinear functions of the embedding of the two merged nodes $(u, v) = m^{(k)}$—similar to the score used in (1) for $q_m$. We use a symmetric form for $\left(\alpha_\text{G}^{(k)}, \lambda_\text{G}^{(k)}\right)$ and an asymmetric one for $\left(\alpha_\text{B}^{(k)}, \beta_\text{B}^{(k)}\right)$. The results of the bilinear forms are passed to a softplus function $\text{softplus}(x) = \log\left(1 + e^x\right)$ to ensure their positivity, and we further add 1 for all parameters but $\lambda_\text{G}^{(k)}$, as they must be larger than 1. We finally obtain the joint probability $q_\ell\left(\ell^{(k)}|m^{(\leq k)}\right)$ of the two branch lengths as $q_s\left(s^{(k)}|m^{(\leq k)}\right) q_r\left(r^{(k)}|m^{(\leq k)}\right)/s^{(k)}$, where the $1/s^{(k)}$ factor arises from the determinant of the Jacobian of the change of variables from $\left(s^{(k)}, r^{(k)}\right)$ to $\ell^{(k)}$ (see Appendix A.2).

Algorithm S.10 summarizes our procedure to evaluate the posterior $q_{\psi(x)}\left(\theta = (\tau, \ell)|x\right)$ of a phylogeny $\tau$ given set of sequences $x$. We use this procedure during the training phase to compute the loss function of Phyloformer 2, and at inference time when we want to evaluate the posterior probability of a phylogeny using a trained network. Of note, when evaluating $q_{\psi(x)}\left(\theta = (\tau, \ell)|x\right)$ the choice of one merge over several possibilities at each step is determined by the data, and does not depend on the network parameters. We therefore don't need to differentiate through discrete operations during training even though the loss evaluation depends on a succession of discrete choices. By contrast, sampling from $q_{\psi(x)}\left(\theta = (\tau, \ell)|x\right)$ does require a sequence of discrete decisions that depend on the network parameters but we never need to differentiate through this process.

Once the network is trained, sampling from the posterior given a set of sequences $x$ (Algorithm S.11) is very similar to the posterior evaluation procedure described above. The tree is iteratively built from evoPF embeddings of $x$ by successively sampling merges from the softmin-parametrized conditional-merge probabilities (1). Similarly, we obtain branch-lengths at a step $k$ by sampling

their sum from the shifted Gamma distribution and their ratio from the Beta distribution, whose parameters are obtained from evoPF embeddings (2). We can also use the trained network to build a greedy approximation of the maximum *a posteriori* (MAP) estimator (thereafter referred to as greedy MAP), by using the mode of the estimated distributions, *i.e.*, replacing the softmin in (1) with $\hat{m}_{\text{greedyMAP}}^{(k)} = \arg\min_{m \in \mathcal{C}^{(k)}} \text{score}_m$ to sample the most probable merge at each step $k$, and using the Gamma and Beta modes for branch lengths, *i.e.* $\hat{s}_{\text{greedyMAP}}^{(k)} = \left(\alpha_{\text{G}}^{(k)} - 1\right)/\lambda_{\text{G}}^{(k)}$ and $\hat{r}_{\text{greedyMAP}}^{(k)} = \left(\alpha_{\text{B}}^{(k)} - 1\right) / \left(\alpha_{\text{B}}^{(k)} + \beta_{\text{B}}^{(k)} - 2\right)$.

Of note, the resulting $q_\psi$ has limited expressivity for two reasons. First, the relative probabilities for merging each of the currently available pairs are computed based on the initial embeddings (*i.e.*, the embeddings of those nodes that have not been chosen thus far are not updated as the recursion proceeds). In the true posterior, on the other hand, the relative posterior probability of being the next smallest cherry for a given pair of nodes in principle depends on previous merges. An update of the vector of embeddings at each step of the recursion could be implemented in a future version, but would be considerably more computationally intensive. Second, we model branch lengths posteriors with specific parametric distributions (Gamma and Beta) whereas the true posterior has no reason to match these analytical forms in general.

# 4 EXPERIMENTS

## 4.1 FASTER AND MORE ACCURATE POINT ESTIMATES OF PHYLOGENIES

We trained Phyloformer 2 (PF2) over a large dataset simulated under similar priors to (Nesterenko et al., 2025). This dataset contains $\approx 1.3 \cdot 10^6$ 50-taxa trees simulated under a rescaled birth-death process—effectively corresponding to the prior $p(\theta)$—and, for each tree $\theta$, an MSA $x$ simulated under the LG+G8 probabilistic model of evolution $p(x|\theta)$ (LG substitution matrix of Le & Gascuel (2008) combined with gamma evolution rates Yang (1994), also see Appendix A.3.1).

The likelihood under this model is tractable, making it a favorable setting for maximum-likelihood methods—FastTree (Price et al., 2010) and IQTREE (Minh et al., 2020) in this experiment. We also include FastME (Lefort et al., 2015), a much faster but less accurate method that only requires to compute likelihoods of branches between pairs of sequences.

The PF2 model was then fine-tuned on tree/MSA pairs with sizes ranging from 10 to 170 taxa, simulated under the same priors as the main training set (see Appendices A.3.1 and A.4). This fine-tuning step is necessary to avoid some overfitting to the number of taxa (see also Figure S.4). All MSAs have the same sequence length since overfitting to that metric does not seem to be an issue (see Figure S.6).

We then inferred greedy MAP trees (see Section 3.3) on a test set of sequence alignments simulated under the same priors as the training set, with 10 to 200 taxa, obtained from Nesterenko et al. (2025). Figure 2a compares all tested methods using the normalized Robinson-Foulds distance between simulated and inferred tree—a classical metric to compare topologies counting the proportion of branches that are present in only one of two phylogenies (Robinson & Foulds, 1981). PF2 is a marked improvement over PF, with better topological accuracy across the whole range of tree/MSA size. For trees with 10 to 175 leaves, it also reconstructs trees with better accuracies than IQTree and FastTree, both state-of-the-art maximum-likelihood tree reconstruction methods. The edge of PF2 against maximum-likelihood methods working under the correct model likely arises from its estimating the posterior distribution using the correct prior—the same tree distribution is used to generate training and test samples—which reduces its variance without creating a bias.

In addition to being more topologically accurate, PF2 is much faster than maximum-likelihood estimators: by one order of magnitude compared to FastTree, two compared to IQTREE. For trees with more than 100 leaves PF2 is faster than PF.

It is also important to note that, although PF2 is still memory intensive compared to maximum-likelihood approaches, it scales better than PF allowing PF2 to infer larger trees, faster, despite having 1000 times more parameters than PF (see Figure 2c).

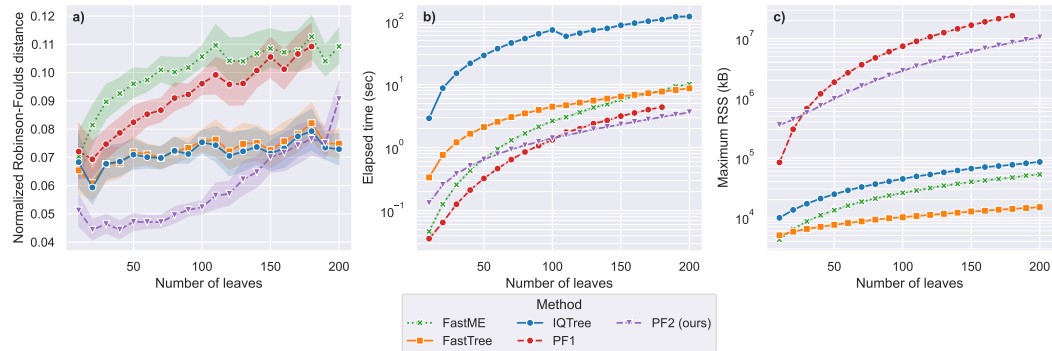

Figure 2: **(a)** Topological performance for Phyloformer 2, measured by the normalized Robinson-Foulds distance. The alignments for which trees were inferred were taken from the original Phyloformer paper (Nesterenko et al., 2025) and were simulated under the LG+GC sequence model. **(b)** Runtime and **(c)** Memory usage for Phyloformer 2. The same GPU model as the original Phyloformer study was used to run Phyloformer 2 inference. Results for other methods are reported from the original Phyloformer paper (Nesterenko et al., 2025).

In order to disentangle the effects of the evoPF module and the BayesNJ loss, we trained a PF2$_{\text{MAE}}$ model using evoPF but replacing the BayesNJ module by a mean absolute error (MAE) loss on pairwise distances, as was done with PF. After a similar number of training steps as PF, we inferred distance matrices from the PF test MSAs, and trees from these matrices using FastME (Lefort et al., 2015). PF2$_{\text{MAE}}$ yields trees with only a slightly better topological accuracy as measured by the RF distance, especially for larger trees (See Figure S.2). This seems to indicate that although the evoPF embedding scheme helps PF2 better predict topologies, most of the topological accuracy gain shown in Figure 2a is due to the BayesNJ loss. Both PF and PF2$_{\text{MAE}}$ yield similar Kuhner-Felsenstein (KF, Kuhner & Felsenstein, 1994) distances—a metric similar to RF but weighting each inconsistent branch by the square of its length.

### 4.2 INCREASED ADVANTAGE UNDER INTRACTABLE PROBABILISTIC MODELS OF EVOLUTION

The main motivation for NPE is to allow well-specified inference under models whose likelihood is intractable. We assess the performance of PF2 in this setting on the same benchmark used in Nesterenko et al. (2025), that includes the Cherry (Prillo et al., 2023) model allowing for dependencies between evolution at distinct positions in the sequences and SelReg model (Duchemin et al., 2022) allowing for heterogeneous distributions between positions. Of note, a mistake in the generation of the Cherry dataset in Nesterenko et al. (2025) inflates the amount of dependency to an unrealistic level (see Appendix A.5) but we keep it as it still provides a proof of concept and to avoid the computational burden of re-training PF on a new dataset. Under both the Cherry and SelReg models, a fine-tuned PF2 model significantly outperforms the equivalent PF model in RF distance and is comparable to PF in terms of KF score (see Figure S.3). For both models and metrics, PF and PF2 outperform all other reference methods.

### 4.3 ESTIMATING THE PHYLOGENETIC POSTERIOR DISTRIBUTION

A major advance of PF2 compared to existing likelihood-free phylogenetic inference methods including PF is its ability to represent entire posterior distributions—as opposed to point estimates—over full phylogenies—as opposed to quartets. We assess the quality of this estimation by comparing against samples obtained from a long MCMC run of RevBayes (Höhna et al., 2016), a standard tool for Bayesian phylogenetic inference. We ran 10 parallel MCMC runs on a single 50 sequence alignment for 50,000 iterations and 5,000 burn-in iterations. We used a uniform prior on tree topologies and an Exponential distribution with $\lambda = 10$ for branch lengths, and LG+G8 as the sequence evolution model. For a fairer comparison, we fine-tuned PF2 on tree/MSA pairs simulated under these priors before sampling from the posterior.

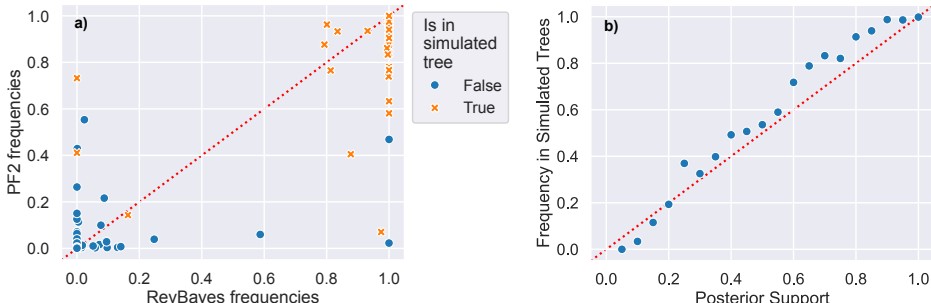

Figure 3: **a)**: Comparison of split frequencies over samples from the posterior of a single 50 sequences MSA, between RevBayes MCMC (x-axis) and PF2 (y-axis). The orange cross marker indicates splits that are present in the tree along which the MSA used for sampling has been simulated. **b)**: simulation-based calibration comparing branch supports, *i.e.*, their frequency in samples from the posterior estimate given by a trained PF2 (x-axis) to the frequency with which splits with a given PF2 support are true, i.e. found in the tree used for simulation (y-axis) Support values are binned by steps of 0.05.

A common way to compare distribution of topologies is through the branches present in sampled trees. Every branch in a phylogeny defines a bipartition of the leaves, making them comparable across all possible trees and providing a softer metric than, *e.g.*, the frequencies of full topologies. Figure 3a shows that RevBayes produces a hard posterior distribution where most branches appear in either all or none of the sampled topologies. PF2 provides a softer posterior but a large agreement with RevBayes, as branches sampled in all RevBayes trees have a frequency larger than 0.6 in PF2, and those not sampled have a frequency mostly lower than 0.3. This is also consistent with our observation that the greedy MAP version of PF2 provides good point estimates, since sequentially sampling the highest probability merges is very likely to select the right branches.

We further investigated the calibration of PF2's posterior probabilities on 196 simulations of trees and alignments with 50 sequences. We computed branch posterior probabilities from samples of $5,000$ trees sampled from PF2 and compared these probabilities to the frequency with which they correspond to branches in the true simulated tree. Figure 3b shows that PF2 is generally well-calibrated, with a tendency to be conservative by generally underestimating branch support. Overall, PF2 appears to provide estimates of posterior tree distributions that should be useful in practice. Furthermore it does so several orders of magnitude faster than MCMC methods like RevBayes by leveraging GPU parallelism (see Appendix A.6).

## 5   CONCLUSION

We introduced Phyloformer 2, a phylogenetic neural posterior estimator combining two novel components: evoPF, an expressive and efficient encoding for aligned sequences, and BayesNJ, a factorization of the tree probability over successive merges. Phyloformer 2 can provide MAP estimates that outperforms existing likelihood-based and -free phylogenetic reconstruction methods while running at least one order of magnitude faster for $PF2_{topo}$. It also approximates posterior distributions in an amortized fashion, with a good calibration, so that branch posterior probabilities provide a useful estimate of their accuracy.

One major limitation of Phyloformer 2 is its scalability, preventing its usage on more than 200 sequences. Future work should explore more efficient encoders (Wohlwend et al., 2025; Wang et al., 2025) or existing heuristics to build larger trees (Warnow, 2018; Jiang et al., 2024). In addition, like other NPE methods, Phyloformer 2 would currently produce poor estimates with no warning when presented with inputs far from its training data, which can happen even when doing well-specified inference—on data generated under the same $p(x, \theta)$ used for training—if some parameter areas have low probability under the prior. Likelihood-based methods are immune to this specific issue, as they do not require a learning phase. By contrast, both likelihood-based and likelihood-free methods perform model-based phylogenetic inference, and as such Phyloformer 2 is not immune to

model mis-specification. Both of these issues could be mitigated by providing an assessment of the uncertainty of its prediction (Gal & Ghahramani, 2016; Lakshminarayanan et al., 2017)

In future work, we will also explore alternative representations for the distribution of trees, such as the ones based on GFlowNet that were successfully leveraged in likelihood-based approaches (Zhou et al.). For the sake of comparison with likelihood-based estimators, the present study mostly focused on tractable probabilistic models, but we expect Phyloformer 2 to reveal most of its potential by allowing inference under more realistic and complex scenarios (Latrille et al., 2021). Other promising directions are to handle unaligned sequences, and inference under broader probabilistic models of evolution embedding phylogenies such as population dynamics and co-evolution.

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

# A   Technical Appendices and Supplementary Material

Technical appendices with additional results, figures, graphs and proofs may be submitted with the paper submission before the full submission deadline (see above), or as a separate PDF in the ZIP file below before the supplementary material deadline. There is no page limit for the technical appendices.

## A.1   Model architecture

**Notation:**   $i$, $j$ and $k$ denote sequence indices, $t$ and $l$ denote residue indices. Capitalized functions (e.g. Linear) are parametrized and learnable, whereas lowercase functions (e.g. sigmoid) are not.

The evoPF module takes as input a multiple sequence alignments $\{\mathbf{x}_{it}\}$, where $\mathbf{x}_{it}$ is the one-hot encoded vector of the $t^{th}$ character of sequence $i$. The evoPF module produces a set sequence embeddings $\{\mathbf{v}_i\}$ of fixed size $c_s = 128$ for each sequence $i$, and a set a sequence pair embeddings $\{\mathbf{z}_{ij}\}$ of fixed size $c_z = 256$ for each pair $(i, j)$ of sequences. Furthermore, the pair embedding vectors $\mathbf{z}_{ij}$ are projected to a single real number $z_{ij}$ to be used as a proxy for phylogenetic distance in the BayesNJ module.

---

**Algorithm S.1** The evoPF module

---

**function** EVOPF($N_{blocs} = 12, \{\mathbf{x}_{it}\}$)
    $\{\mathbf{v}_{it}\}$ = MSAEmbedder($\{\mathbf{x}_{it}\}$)
    $\{\mathbf{z}_{ij}\}$ = PairEmbedder($\{\mathbf{x}_{it}\}$)

    **for all** $l \in [1, \ldots, N_{blocs}]$ **do**
        $\{\mathbf{v}_{it}\}$ += MSAColAttentionWithPairBias($\{\mathbf{v}_{it}\}, \{\mathbf{z}_{ij}\}$)
        $\{\mathbf{v}_{it}\}$ += MSARowAttention($\{\mathbf{v}_{it}\}$)
        $\{\mathbf{v}_{it}\}$ += MSATransition($\{\mathbf{v}_{it}\}$)

        $\{\mathbf{z}_{ij}\}$ += OuterProductMean($\{\mathbf{v}_{it}\}$)

        $\{\mathbf{z}_{ij}\}$ += PairAttention($\{\mathbf{z}_{ij}\}$)
        $\{\mathbf{z}_{ij}\}$ += PairTransition($\{\mathbf{z}_{ij}\}$)

    $z_{ij}$ = Linear($\mathbf{z}_{ij}$)
    $\mathbf{v}_i$ = mean$_t(\mathbf{v}_{it})$
    **return** $\{z_{ij}\}, \{\mathbf{v}_i\}$             $\mathbf{z}_{ij} \in \mathbb{R}^{c_z}, \mathbf{v}_i \in \mathbb{R}^{c_s}$

---

### A.1.1 EMBEDDING MODULES

---

**Algorithm S.2** MSA Embedding module

---

**function** MSAEMBEDDER($\{\mathbf{x}_{it}\}, c_s = 128$)
    $\mathbf{a}_{it}$ = Conv2D($\mathbf{x}_{it}$)             $a_{it} \in \mathbb{R}^{c_s}$
    $\mathbf{v}_{it}$ = reLU($\mathbf{a}_{it}$)
    **return** $\{\mathbf{v}_{it}\}$

---

**Algorithm S.3** Pair Embedding module

---

**function** PAIREMBEDDER($\{\mathbf{x}_{it}\}, c_z = 256$)
    $a_{it}$ = reLU(Conv2D($\mathbf{x}_{it}$))             $a_{it} \in \mathbb{R}^{c_z}$
    $\mathbf{z}_{ij}$ = mean$_t(a_{it} + a_{jt})$             $1 \le j < i \le N$
    **return** $\{\mathbf{z}_{ij}\}$

---

### A.1.2 THE EVOPF MSA STACK

---

**Algorithm S.4** MSA Stack - Column-wise pair-biased gated self-attention

---

**function** MSACOLATTENTIONWITHPAIRBIAS($\{\mathbf{v}_{it}\}, \{\mathbf{z}_{ij}\}, N_{head} = 4, c = c_s/N_{head}$)
    $\mathbf{v}_{it} \leftarrow$ LayerNorm($\mathbf{v}_{it}$)
    $\mathbf{q}_{it}^h, \mathbf{t}_{it}^h, \mathbf{v}_{it}^h,$ = LinearNoBias($\mathbf{v}_{it}$)      $\mathbf{q}_{it}^h, \mathbf{t}_{it}^h, \mathbf{v}_{it}^h \in \mathbb{R}^c, 1 \le h \le N_{head}$
    $b_{ij}^h$ = LinearNoBias($\mathbf{z}_{ij}$)
    $\mathbf{g}_{it}^h$ = sigmoid(Linear($\mathbf{v}_{it}$))             $\mathbf{g}_{it} \in \mathbb{R}^c$

    $a_{ijt}^h$ = softmax $\left( \frac{1}{\sqrt{c}} \mathbf{q}_{it}^{h\top} \mathbf{t}_{jt}^h + b_{ij}^h \right)$
    $\mathbf{o}_{it}^h = \mathbf{g}_{it}^h \odot \sum_j a_{ijt}^h \mathbf{v}_{it}^h$

    $\tilde{\mathbf{s}}_{it}$ = Linear $\left( \text{concat}_h(\mathbf{o}_{it}^h) \right)$             $\tilde{\mathbf{s}}_{it} \in \mathbb{R}^{c_s}$
    **return** $\{\tilde{\mathbf{s}}_{it}\}$

---

---

**Algorithm S.5** MSA Stack - Row-wise gated self-attention

---

**function** MSAROWATTENTION($\{\mathbf{v}_{it}\}, N_{head} = 4, c = c_s/N_{head}$)

$\quad \mathbf{v}_{it} \leftarrow \text{LayerNorm}(\mathbf{v}_{it})$

$\quad \mathbf{q}_{it}^h, \mathbf{t}_{it}^h, \mathbf{v}_{it}^h, = \text{LinearNoBias}(\mathbf{v}_{it})$ $\qquad\qquad\qquad$ $\mathbf{q}_{it}^h, \mathbf{t}_{it}^h, \mathbf{v}_{it}^h \in \mathbb{R}^c, 1 \leq h \leq N_{head}$

$\quad \mathbf{g}_{it}^h = \text{sigmoid}(\text{Linear}(\mathbf{v}_{it}))$ $\qquad\qquad\qquad\qquad\qquad\qquad\qquad$ $\mathbf{g}_{it} \in \mathbb{R}^c$

$\quad a_{itl}^h = \text{softmax}\left(\frac{1}{\sqrt{c}}\mathbf{q}_{it}^{h\top}\mathbf{t}_{il}^h\right)$

$\quad \mathbf{o}_{it}^h = \mathbf{g}_{it}^h \odot \sum_l a_{itl}^h \mathbf{v}_{il}^h$

$\quad \tilde{\mathbf{s}}_{it} = \text{Linear}\left(\text{concat}_h(\mathbf{o}_{it}^h)\right)$ $\qquad\qquad\qquad\qquad\qquad\qquad\qquad$ $\tilde{\mathbf{s}}_{it} \in \mathbb{R}^{c_s}$

$\quad$ **return** $\{\tilde{\mathbf{s}}_{it}\}$

---

**Algorithm S.6** MSA Stack - Transition

---

**function** MSATRANSITION($\{\mathbf{v}_{it}\}, n = 4$)

$\quad \mathbf{v}_{it} \leftarrow \text{LayerNorm}(\mathbf{v}_{it})$

$\quad \mathbf{a}_{it} = \text{Linear}(\mathbf{v}_{it})$ $\qquad\qquad\qquad\qquad\qquad\qquad\qquad\qquad\qquad$ $\mathbf{a}_{it} \in \mathbb{R}^{nc_s}$

$\quad \mathbf{v}_{it} \leftarrow \text{Linear}(\text{reLU}(\mathbf{a}_{it}))$

$\quad$ **return** $\{\mathbf{v}_{it}\}$

---

**Algorithm S.7** Communication - Outer product mean

---

**function** OUTERPRODUCTMEAN($\{\mathbf{v}_{ik}\}, c = 32$)

$\quad \mathbf{v}_{it} \leftarrow \text{LayerNorm}(\mathbf{v}_{it})$

$\quad \mathbf{a}_{it}, \mathbf{b}_{it} = \text{Linear}(\mathbf{v}_{it})$ $\qquad\qquad\qquad\qquad\qquad\qquad\qquad$ $\mathbf{a}_{it}, \mathbf{b}_{it} \in \mathbb{R}^{nc_s}$

$\quad \mathbf{o}_{ij} \leftarrow \text{flatten}(\text{mean}_t(\mathbf{a}_{it} \otimes \mathbf{b}_{it}))$ $\qquad\qquad\qquad\qquad\qquad\qquad$ $\mathbf{o}_{ij} \in \mathbb{R}^{c^2}$

$\quad \mathbf{z}_{ij} = \text{Linear}(\mathbf{o}_{ij})$ $\qquad\qquad\qquad\qquad\qquad\qquad\qquad\qquad\qquad$ $\mathbf{z}_{ij} \in \mathbb{R}^{c_z}$

$\quad$ **return** $\{\mathbf{z}_{ij}\}$

---

### A.1.3 THE EVOPF PAIR STACK

---

**Algorithm S.8** Pair Stack - Gated self-attention

---

**function** PAIRATTENTION($\{\mathbf{z}_{ij}\}, N_{head} = 4, c = c_z/N_{head}$)

$\quad \mathbf{z}_{ij} \leftarrow \text{LayerNorm}(\mathbf{z}_{ij})$

$\quad \mathbf{q}_{ij}^h, \mathbf{k}_{ij}^h, \mathbf{v}_{ij}^h, = \text{LinearNoBias}(\mathbf{z}_{ij})$ $\qquad\qquad\qquad$ $\mathbf{q}_{ik}^h, \mathbf{k}_{ij}^h, \mathbf{v}_{ij}^h \in \mathbb{R}^c, 1 \leq h \leq N_{head}$

$\quad \mathbf{g}_{ij}^h = \text{sigmoid}(\text{Linear}(\mathbf{z}_{ij}))$ $\qquad\qquad\qquad\qquad\qquad\qquad\qquad$ $\mathbf{g}_{ij} \in \mathbb{R}^c$

$\quad a_{ijk}^h = \text{softmax}\left(\frac{1}{\sqrt{c}}\mathbf{q}_{ij}^{h\top}\mathbf{k}_{jk}^h\right)$

$\quad \mathbf{o}_{ij}^h = \mathbf{g}_{ij}^h \odot \sum_k a_{ijk}^h \mathbf{v}_{ik}^h$

$\quad \tilde{\mathbf{z}}_{ij} = \text{Linear}\left(\text{concat}_h(\mathbf{o}_{ij}^h)\right)$ $\qquad\qquad\qquad\qquad\qquad\qquad\qquad$ $\tilde{\mathbf{z}}_{ik} \in \mathbb{R}^{c_z}$

$\quad$ **return** $\{\tilde{\mathbf{z}}_{ij}\}$

---

**Algorithm S.9** Pair Stack - Transition

---

**function** PAIRTRANSITION($\{\mathbf{z}_{ij}\}, n = 4$)

$\quad \mathbf{z}_{ij} \leftarrow \text{LayerNorm}(\mathbf{z}_{ij})$

$\quad \mathbf{a}_{ij} = \text{Linear}(\mathbf{z}_{ij})$ $\qquad\qquad\qquad\qquad\qquad\qquad\qquad\qquad\qquad$ $\mathbf{a}_{ij} \in \mathbb{R}^{nc_z}$

$\quad \mathbf{z}_{ij} \leftarrow \text{Linear}(\text{reLU}(\mathbf{a}_{ij}))$

$\quad$ **return** $\{\mathbf{z}_{ij}\}$

---

### A.1.4 Evaluating $q_{\psi(x)}\left(\theta = (\tau, \ell)|x\right)$ with BayesNJ

---

**Algorithm S.10** The BayesNJ Loss

---

**function** BayesNJ(Embeddings $\{\psi(x)\} = \{\mathbf{v}_1, \ldots, \mathbf{v}_N\}$, merges $\{m^{(k)}\}$, branch lengths $\{\ell^{(k)}\}$)

$\quad \mathcal{L}_\tau \leftarrow 0, \mathcal{L}_\ell \leftarrow 0, \{c_m^{(1)}\} \leftarrow \{0 \,\forall m \in \mathcal{S}_0^2\}$      *Initialize log probabilities and constraints*

$\quad$ **for all** $k \in [1, \ldots, N-2]$ **do**

$\quad\quad$ # Topological component

$\quad\quad \{\text{score}_{m=(u,v)}\} \leftarrow \{\text{SymmetricBilinear}(\mathbf{v}_u, \mathbf{v}_v)\}$      *Score all pairs $m = (u, v) \in \mathcal{C}(k)$*

$\quad\quad \{q_m(m)\} \leftarrow \text{softmin}(\{\text{score}_m\})$

$\quad\quad \mathcal{L}_\tau \mathrel{+}= \log(q_m(m^{(k)}))$

$\quad\quad$ # Branch length component

$\quad\quad \mathbf{v}_{N+k} \leftarrow \text{SymmetricBilinear}(m^{(k)})$      *Compute parent embedding*

$\quad\quad \left(\tilde{\alpha}_G^{(k)}, \tilde{\lambda}_G^{(k)}\right) \leftarrow \text{SymmetricBilinear}(m^{(k)})$

$\quad\quad \tilde{\alpha}_G^{(k)} \leftarrow 1 + \tilde{\alpha}_G^{(k)}$

$\quad\quad \left(\alpha_G^{(k)}, \lambda_G^{(k)}\right) \leftarrow \text{softplus}\left(\tilde{\alpha}_G^{(k)}, \tilde{\lambda}_G^{(k)}\right) + \varepsilon$

$\quad\quad \left(\tilde{\alpha}_B^{(k)}, \tilde{\beta}_B^{(k)}\right) \leftarrow \text{Bilinear}(m^{(k)})$

$\quad\quad \left(\alpha_B^{(k)}, \beta_B^{(k)}\right) \leftarrow 1 + \text{softplus}\left(\tilde{\alpha}_B^{(k)}, \tilde{\beta}_B^{(k)}\right) + \varepsilon$

$\quad\quad \mathcal{L}_\ell \mathrel{+}= \log \text{PDF}_{\text{Gamma}}\left(\ell_i^{(k)} + \ell_j^{(k)} - c_{m^{(k)}}^{(k)}|\alpha_G^{(k)}, \lambda_G^{(k)}\right) +$

$\quad\quad \log \text{PDF}_{\text{Beta}}\left(\ell_i^{(k)}/\ell_j^{(k)}|\alpha_B^{(k)}, \beta_B^{(k)}\right)$

$\quad\quad \{c_m^{(k+1)}\} \leftarrow \left\{\max\left(c_m^{(k)}, \ell_i^{(k)} + \ell_j^{(k)}\right), \forall m \in \mathcal{S}_{(k)}\right\}$      *Update constraints*

$\quad\quad$ # Prepare next iteration

$\quad\quad \mathcal{S}_{(k+1)} = \{\mathcal{S}_{(k)} \cup u\} \setminus m^{(k)}$      *Update mergeable nodes*

$\quad$ **return** $\mathcal{L}_\tau, \mathcal{L}_\ell$

---

A.1.5 SAMPLING FROM $q_{\psi(x)}\left(\theta = (\tau, \ell)|x\right)$ WITH BAYESNJ

---

**Algorithm S.11** Sampling from the posterior

For the greedy MAP approximation, we simply replace the softmin of scores with an argmin to sample merges, and replace samplings from Gamma and Beta distributions with the corresponding mode.

**function** SAMPLINGBAYESNJ(Embeddings $\{\psi(x)\} = \{\mathbf{v}_1, \ldots, \mathbf{v}_N\}$)

   $\{c_{ij}^{(1)}\} \leftarrow \{0, \forall (i, j) \in [1, \ldots, N-2]^2\}$           *Initialize constraints*

   **for all** $k \in [1, \ldots, N-2]$ **do**

      # Topological component

      $\{\text{score}_{ij}\} \leftarrow \{\text{SymmetricBilinear}(\mathbf{v}_i, \mathbf{v}_j)\}$       *Score all pairs $(i, j) \in \mathcal{S}_{(k)}^2$*

      $m^{(k)} \sim \text{softmin}(\{\text{score}_{ij}\})$                   *Sample merge*

      # Branch length component

      $\mathbf{v}_{N+k} \leftarrow \text{SymmetricBilinear}(m^{(k)})$       *Compute parent embedding*

      $\left(\tilde{\alpha}_{\text{G}}^{(k)}, \tilde{\lambda}_{\text{G}}^{(k)}\right) \leftarrow \text{SymmetricBilinear}(m^{(k)})$

      $\tilde{\alpha}_{\text{G}}^{(k)} \leftarrow 1 + \tilde{\alpha}_{\text{G}}^{(k)}$

      $\left(\alpha_{\text{G}}^{(k)}, \lambda_{\text{G}}^{(k)}\right) \leftarrow \text{softplus}\left(\tilde{\alpha}_{\text{G}}^{(k)}, \tilde{\lambda}_{\text{G}}^{(k)}\right) + \varepsilon$

      $\left(\tilde{\alpha}_{\text{B}}^{(k)}, \tilde{\beta}_{\text{B}}^{(k)}\right) \leftarrow \text{Bilinear}(m^{(k)})$

      $\left(\alpha_{\text{B}}^{(k)}, \beta_{\text{B}}^{(k)}\right) \leftarrow 1 + \text{softplus}\left(\tilde{\alpha}_{\text{B}}^{(k)}, \tilde{\beta}_{\text{B}}^{(k)}\right) + \varepsilon$

      $s^{(k)} \sim c_{m^{(k)}}^{(k)} + \text{Gamma}(\alpha_{\text{G}}^{(k)}, \lambda_{\text{G}}^{(k)})$              *Sample sum*

      $r^{(k)} \sim \text{Beta}(\alpha_{\text{B}}^{(k)}, \beta_{\text{B}}^{(k)})$                  *Sample ratio*

      $\ell_i^{(k)} \leftarrow r^{(k)} \cdot s^{(k)}$

      $\ell_j^{(k)} \leftarrow s^{(k)} - \ell_j^{(k)}$

      # Prepare next iteration

      $\{c_{vw}^{(k+1)}\} \leftarrow \left\{\max\left(c_{vw}^{(k)}, s^{(k)}\right), \forall (v, w) \in \mathcal{S}_{(k)}\right\}$    *Update constraints*

      $\mathcal{S}_{(k+1)} = \{\mathcal{S}_{(k)} \cup u\} \setminus m^{(k)}$        *Update mergeable nodes*

   **return** $\mathcal{L}_\tau, \mathcal{L}_\ell$

---

A.2 REPARAMETERIZATION OF THE BRANCH LENGTH PROBABILITY $q_\ell$

Because we want to constrain the sum of branch lengths in our probability distribution, we reparameterize $\ell = (\ell_i, \ell_j)$ as $(s, r) = g(\ell) = (\ell_i + \ell_j, \ell_i/(\ell_i + \ell_j))$ and model the distribution of $q_{(s,r)}(s, r) = q_s(s)q_r(r)$, as detailed in 3.3. In order to recover the probability $q_\ell$ from $q_s(s)q_r(r)$, we must account for this change of variable:

$$q_\ell(\ell) = q_{(s,r)}\left(g(\ell)\right) |\det J_g(\ell)|,$$

where

$$J_g(\ell) = \begin{pmatrix} 1 & 1 \\ \frac{\ell_j}{\ell_i + \ell_j} & -\frac{\ell_i}{\ell_i + \ell_j} \end{pmatrix}$$

is the Jacobian matrix of $g$, whose determinant is therefore $-1/(\ell_i + \ell_j) = -1/r$. As a result,

$$q_\ell(\ell) = q_s(s)q_r(r)/r.$$

## A.3 TRAINING RUNS

| Model | Starting weights | Embedding dimensions $(c_s\|c_z)$ | Training data | Loss function | Training time | Batch size | Scheduled epochs | Target LR | Warmup | Selected step |
|---|---|---|---|---|---|---|---|---|---|---|
| (PF2) | Random | $(128\|256)$ | BD,LG+G8 | BayesNJ | 62.5h | 16 | 30 | $10^{-4}$ | $10^3$ | 86000 |
| **PF2**$_{MAE}$ | Random | $(128\|256)$ | BD,LG+GC | MAE | 26h | 16 | 30 | $5 \cdot 10^{-4}$ | $10^3$ | 51000 |
| **PF2** | (PF2) | $(128\|256)$ | BD,LG+G8,multi | BayesNJ | 8.5h | 1-40 | 30 | $10^{-6}$ | 0.5% | 8000 |
| **PF2**$_{Cherry}$ | (PF2) | $(256\|512)$ | BD,Cherry | BayesNJ | 42h | 6 | 30 | $10^{-5}$ | 0.5% | 41000 |
| **PF2**$_{SelReg}$ | (PF2) | $(256\|512)$ | BD,SelReg | BayesNJ | 42h | 6 | 30 | $10^{-5}$ | 0.5% | 44000 |
| **PF2**$_{MCMC}$ | (PF2) | $(256\|512)$ | U+Exp,LG+G8 | BayesNJ | 10h | 6 | 30 | $10^{-6}$ | 0.5% | 12009 |

Table S.1: Training run parameters. All runs were in a distributed data-parallel setting using 4 H100 GPUs. Final models are shown with their name in bold. More details on the training data in Appendix A.3.1 and Figure S.1.

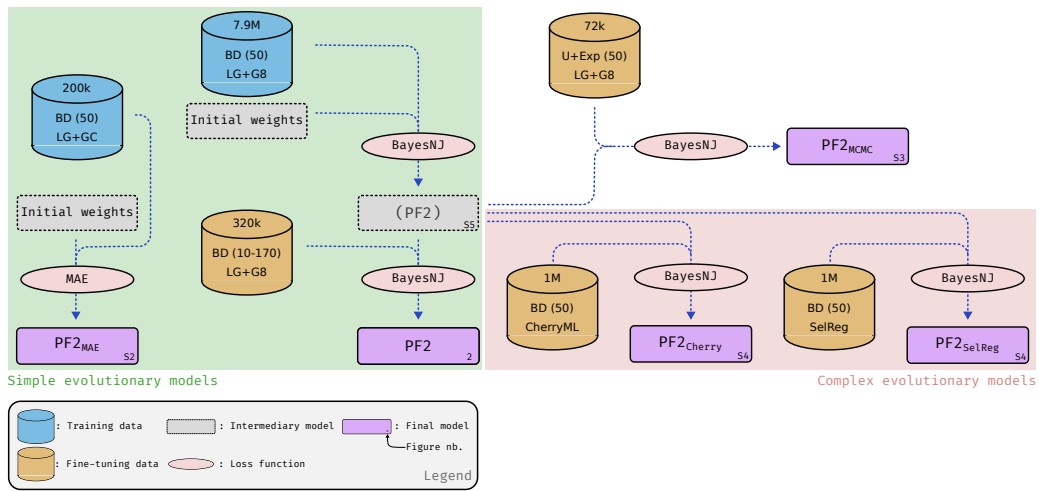

Figure S.1: Training setting for all PF2 instances. PF2 instances are shown in rectangles, with a full outline and purple fill if they are final models used for inference in the main results, and a dotted outline and gray fill if they are used as the starting model for fine tuning runs. The loss functions used for each training or fine-tuning runs are shown in ovals. Finally the training (blue) and fine-tuning (yellow) datasets are also shown as cylindrical shapes. For each dataset, the number of training and validation examples is shown in the top section of the cylinder. The simulation priors are shown in the body of each cylinder : (1) the tree topology prior with the training tree size in parentheses, (2) the MSA evolutionary model. The tree priors are either *(BD)*: rescaled Birth-death, described in (Nesterenko et al., 2025), or *(U+Exp)*: Uniform tree topology with $\lambda = 10$ exponentially distributed branch lengths available in RevBayes (Höhna et al., 2016). The number of figure in which the performance of a given model is studied is shown in the bottom right corner of the corresponding rectangle. Datasets are described in Appendix A.3.1

### A.3.1 TRAINING DATASETS

**From (Nesterenko et al., 2025):** The 200k LG+GC, Cherry and Selreg datasets, used for training and fine-tuning models were obtained from the original paper. A full description of these datasets is available in the supplementary material of (Nesterenko et al., 2025)

**LG+G8 dataset:** This is the main dataset used to train PF2. the 7,966,499 training trees (and 10,000 validation trees) are simulated with 50 leaves, following the empirically rescaled birth-death procedure described in (Nesterenko et al., 2025). MSA were simulated along these trees using

IQTree's alisim tool (Minh et al., 2020) under the LG evolutionary model. Rate heterogeneity across sites was modeled with an 8 category discrete Gamma. Insertions-deletion events were added using alisim, with identical rates of $2 \cdot 10^{-3}$, and insertion-deletion length was modeled with the default options: a zipfian distribution with an exponent of 1.7 and a maximum size of 100. These parameters were chosen to yield MSAs with 10% of gaps on average on trees sampled from our prior distribution, which seems consistent with the gap content of empirical data.

**Multi-size LG+G8 dataset:** This dataset was used to fine-tune PF2 to limit the effect of overfitting to the training. Tree-MSA pairs were generated with 10 to 170 leaves (with a step-size of 10) using the same procedure as the main LG+G8 dataset described above. We simulated 20,000 training examples and 1,000 validation examples for each tree size.

**MCMC fine-tuning dataset:** This dataset was used to fine tune PF2 under the prior used in RevBayes (Höhna et al., 2016) to estimate the posterior distribution shown in Figure 3. 72,007 training (resp. 1000 validation) trees were simulated with RevBayes with uniformly distributed topologies and branch lengths sampled from an exponential distribution with $\lambda = 10$. MSAs were simulated along those trees using the same procedure as the main LG+G8 and the multi-size LG+G8 datasets described above.

## A.4 FINE TUNING ON DIFFERENT TREE-SIZES

In our current implementation, the BayesNJ module only accepts batches with MSA with the same number of taxa. In order to avoid wasting GPU memory we use a dynamic batch-size during fine-tuning, where the number of examples for a batch with MSAs of a certain size is set to use as much GPU memory as it can. Since batch-size and learning rate are tightly linked, we define the learning-rate schedule for MSAs with 50 sequences and rescale the learning rate applied at a certain batch by the ratio of the current batch-size and the one for MSAs with 50 sequences. We also set a maximum batch-size equal to the one chosen for MSAs of 50 sequences, this ensures that the model sees enough batches for small MSAs during fine-tuning.

## A.5 THE CHERRY DATASET IN NESTERENKO ET AL. (2025) CONTAINS UNREALISTIC AMOUNTS OF COEVOLUTION BETWEEN SITES

The Cherry dataset simulated in Nesterenko et al. (2025) was simulated by using an incorrect rate matrix. As a result, the amount of co-evolution among pairs of coevolving residues was superior to what can be found in empirical data. The cause of this high amount of coevolution is a mistake in the use of the Cherry matrix Prillo et al. (2023). Instead of using the Cherry rate matrix to model pairs of interacting sites, the authors used the product of the rate matrix with its stationary frequencies. In standard models of molecular evolution, it is customary to represent a reversible substitution rate matrix $Q$ as a product of an exchangeability matrix $R$ and stationary frequencies $F$: $Q = R \times F$. The Cherry dataset in Nesterenko et al. (2025) was actually simulated according to a matrix $Q' = R \times F \times F = Q \times F$. The resulting data provides an example of data with extreme amounts of coevolution, which we use as an example of strong departure from standard models of sequence evolution as implemented in e.g., IQ-Tree Minh et al. (2020).

## A.6 BREAK EVEN ANALYSIS

We analyze the total compute time needed to obtain inferred trees for PF2 and ML methods in Table S.2. While PF2's training time on 7.9M (tree,alignment) pairs was 62h on GPUs, the simulation of the 7.9M pairs took approximately 1450h of CPU time. Similarly, PF2's fine-tuning time on $16 \times 20$k (tree,alignment) pairs was 8h30 on GPUs while the simulation of these pairs took 288 CPU hours. This is because we discard all alignments that contain identical sequences, which forces us to simulate many more pairs ($6\times$ more for MSAs with 50 sequences, and up to $60\times$ more for alignments with 170 tips) than we actually use. Other simulation protocols may be much shorter, and we plan to improve this step of our pipeline. In the following, we report break-even points both with and without the simulation. Because of this cost, if we only infer trees for MSAs of 50 tips, PF2 would break even in terms of compute time after 230k tree inferences with IQTree with the LG+GC model (9042 tree inferences without simulation), 16k with IQTree and ModelFinder (629), or 4.3M

FastTree inferences (169047). If we focus on trees of 100 tips this goes down to 91k IQTree with LG+GC (3577), 8k IQTree with ModelFinder (315) and 2.2M FastTree inferences (86489). While the break-even points are quite high for point estimation, this changes radically when sampling a large number of trees from the learned posterior distribution. Since PF2 can sample from the posterior in parallel using GPUs it is very fast, and for trees with 50 tips we reach the break even point w.r.t RevBayes in only 11 sampling runs of 10k iterations. In practice it might be even lower because some of the MCMC samples must be discarded during the burn-in phase.

| | Simulation | Training | MAP ($n = 50$) | MAP ($n = 100$) | 10k Samples ($n = 50$) |
|---|---|---|---|---|---|
| **PF2** | 1735h | 71h | 0.6 | 1.4 | 38.8 |
| IQTree | - | - | 28.9 | 73.1 | - |
| IQTree+MF | - | - | 405.5 | 826.6 | - |
| FastTree | - | - | 2.1 | 4.4 | - |
| RevBayes | - | - | - | - | 172h |

Table S.2: Time per inference and upfront tasks for PF2 vs ML and MCMC methods. Simulation times are estimated from Snakemake pipeline reports and are reported in CPU time. In practice this is highly parallelisable so wall time would be much shorter. Here we compare PF2 and other methods on 2 tasks. (1) Point estimations (for 50-sequence and 100-sequence MSAs) where we compare ourselves to maximum-likelihood methods: IQTree, IQTree with ModelFinder and FastTree, and (2) Sampling 10k trees from the posterior for a 50-sequence MSA, where we compare ourselves to RevBayes. Times for maximum likelihood methods are obtained from (Nesterenko et al., 2025). The mean time to sample 10k trees from the posterior distribution, replicated over 50 50-sequence MSAs, using PF2 is reported. PF2 samples from the posterior in batches of 500 parallel samples. For the RevBayes estimate, we measured how many trees were able to be sampled during a fixed 20h duration and extrapolate. Simulation and Training time for PF2 are the sum of the times for the main training and fine-tuning phases. Reported times are in seconds unless specified otherwise.

## A.7 SUPPLEMENTARY FIGURES

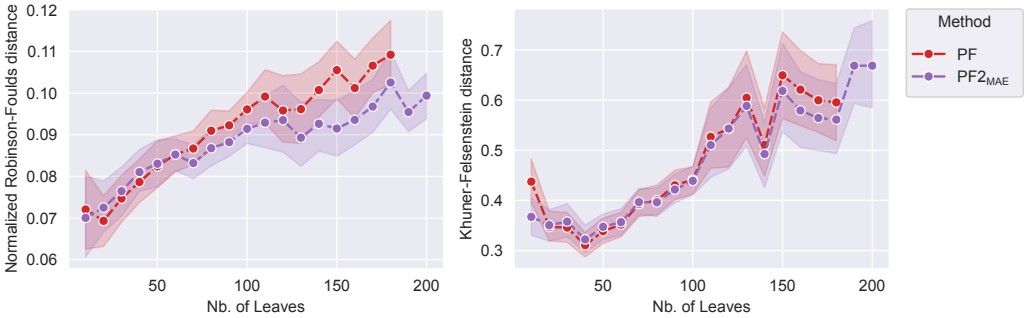

Figure S.2: How does PF2 trained with an L1 loss compare to PF? PF2 was trained under the same conditions as the original PF on LGGC trees with an L1 loss on pairwise phylogenetic distances. The Robinson-Foulds distance (left) shows the topological reconstruction accuracy, while the Kuhner-Felsenstein distance (right) takes both topology and branch-lengths into account.

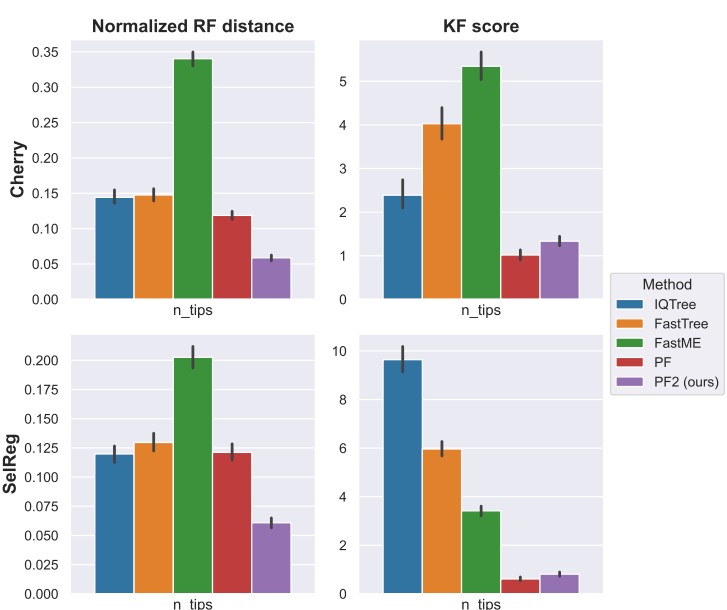

Figure S.3: Average PF2 performance under intractable-likelihood models, measured on 50-tip trees. The PF and PF2 versions are fine tuned to either Cherry (top row) or the SelReg (bottom row) data. Error bar show the 95% CI computed with 1000 bootstrap samples.

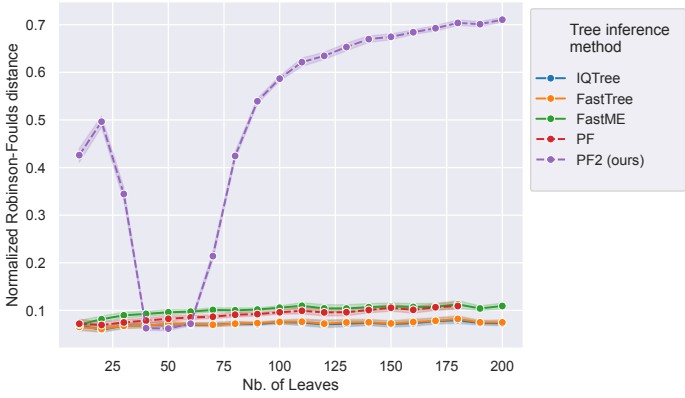

Figure S.4: Topological performance for non fine-tuned topology-only Phyloformer 2, measured by the normalized Robinson-Foulds distance. The MSA dataset and compared method results are the same as Figure 2

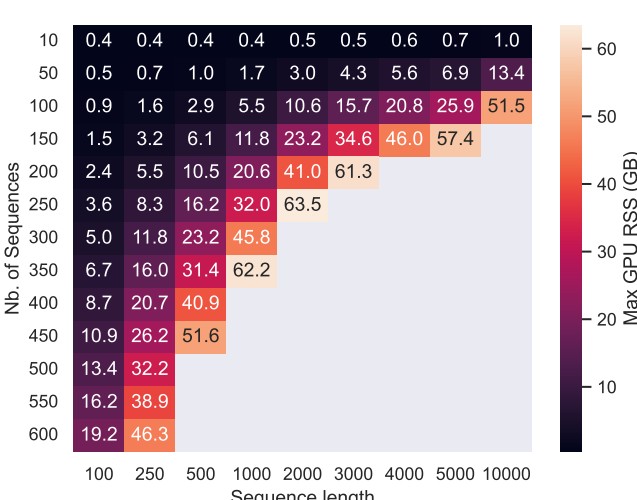

Figure S.5: PF2 memory usage (in GB) scaling w.r.t number of sequences and sequence length measured on an H100 GPU

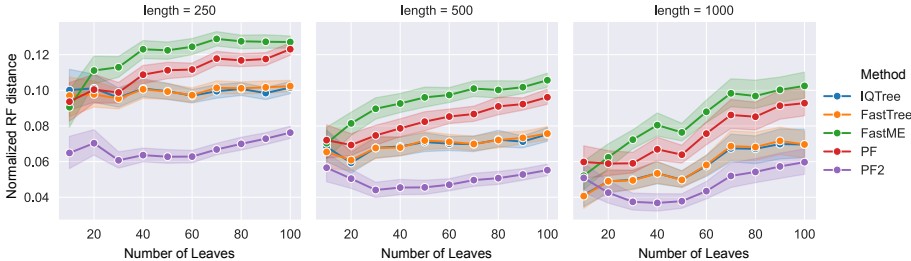

Figure S.6: Topological reconstruction accuracy for PF2 and other methods for different sequence lengths. MSAs of different lengths are simulated on the same set of phylogenetic trees for the 3 panels.

