# OpenReview forum: "Likelihood-free inference of phylogenetic tree posterior distributions"
_ICLR.cc/2026/Conference — Submitted to ICLR 2026_

### Official Review · Reviewer_7J8h · 2025-10-17

**Soundness:** 2
**Presentation:** 4
**Contribution:** 3
**Rating:** 8
**Confidence:** 3

**Summary:**

The paper introduces Phyloformer 2, a novel deep learning framework for likelihood-free phylogenetic inference. This method directly estimates posterior distributions over phylogenetic tree topologies from multiple sequence alignments (MSAs). The model's architecture integrates two key components: EvoPF, an EvoFormer-inspired encoder that extracts informative pairwise and sequence-level embeddings, and BayesNJ, a neural posterior estimator that defines a probabilistic factorization over successive subtree merges.
The authors demonstrate that Phyloformer 2 outperforms traditional likelihood-based methods, such as IQ-TREE and FastTree, as well as the previous likelihood-free model, Phyloformer. Notably, it achieves this with a significant speedup of up to two orders of magnitude. A key advancement of this work is its ability to provide approximate posterior distributions, moving beyond the point estimates offered by many existing methods. By extending neural posterior estimation (NPE) to entire trees rather than just quartets, this paper presents a significant conceptual and methodological step forward for simulation-based inference in evolutionary genomics. The manuscript is clearly written and well-organized, with detailed explanations of the model architecture, training procedures, and evaluation protocols.
Major Concerns

**Strengths:**

The method delivers clear advantages in both accuracy and efficiency, outperforming state-of-the-art likelihood-based and neural network approaches while running orders of magnitude faster. The paper is clearly written, well-structured, and demonstrating the approach’s potential to substantially advance simulation-based inference using genomic data.

**Weaknesses:**

1. Despite the promising results, I have several concerns regarding the practical applicability and scalability of the proposed method.
Generalizability to Longer Sequences: The model was trained and evaluated exclusively on sequences with a maximum length of 500 base pairs. It is unclear how the model's accuracy would be affected by longer alignments, which are common in phylogenomic studies. An analysis of the relationship between sequence length and performance is needed to assess the model's utility for more complex datasets.
2. Scalability with an Increasing Number of Taxa: As shown in Figure 2, the model's error rate increases drastically when the number of taxa exceeds 100. This sharp decline in accuracy raises questions about the practical utility of Phyloformer 2 for the larger datasets frequently encountered in modern phylogenetic analyses.
3. Lack of Validation on Empirical Data: The method was not applied to any empirical datasets. While performance on simulated data is strong, its effectiveness on real-world biological data—with all its inherent complexities and noise—remains untested. Validating the model on well-established empirical benchmarks is crucial to demonstrate its practical relevance.
4. Implementation Limitations: The current implementation is limited to a maximum of 200 sequences. This constraint severely restricts its applicability for many research questions, which often involve hundreds or even thousands of taxa.

**Questions:**

Could the authors discuss the challenges they anticipate in applying Phyloformer 2 to empirical datasets, which often contain missing data, duplication loss, etc?

---

> ### Author Response · Authors · 2025-11-21
> **Author response to reviewer 3**
>
> We thank the reviewer for their helpful comments. Please find our answer below:
>
> ## Weaknesses
>
>  - **(1)** We agree that this is a valid concern, and we have added supplementary figure S6 and referenced it in the text *(see lines 360-361)*. Figure S6 shows that changing the sequence length impacts performance similarly across all tested methods, indicating no significant overfitting to sequence length.
>  - **(2)** We agree that the sharp increase in RF distance for large trees is concerning. We believe it is due to the fact that during fine-tuning the model has not been trained on a large enough number of these big trees. We are currently training and fine-tuning a new version of PF2 to address this concern. We will include these results in our following revision.
>  - **(3)** Again, this is also something that we will address in our following revision.
>  - **(4)** While Phyloformer 2's scaling is not necessarily adapted to building very large phylogenetic trees, we stress that Phyloformer 2 is an important conceptual step and a tangible proof of concept that it is possible to do likelihood-free NPE for intractable evolutionary models. In the discussion we mention approaches to improve the scaling and out-of-distribution confidence checks, both of which are ultimately needed to make this practically usable, however we consider this to be out of scope for this current work.  Please also see our answer to reviewer 1's weakness 3.
>
> ## Questions
>
> We aim to address these concerns in our following revision.

---

### Official Review · Reviewer_Jzst · 2025-10-30

**Soundness:** 2
**Presentation:** 1
**Contribution:** 2
**Rating:** 2
**Confidence:** 4

**Summary:**

The submission tackles an exciting and important problem in phylogenetic inference, namely solving the computational complexity induced by the likelihood function. By considering a likelihood-free learning approach, the submission contains the proposition of Phyloformer 2 which uses an attention network (EvoPF) to encode MSA data to the parameters of an approximate posterior distribution. Phyloformer 2 shows promising inference time results compared to baselines on synthetic datasets.

**Strengths:**

The significance of the problem that this submission aims to solve is very big. Further, the authors cleverly modify the Evoformer architecture from the AlphaFold 2 paper in order to handle MSA data, resulting in the EvoPF. Also, the idea to do likelihood-free posterior inference is an interesting approach to probabilistic phylogenetic posterior inference -- a topic of interest in the ML community. Additionally Fig. 1 is ambitiously produced and adds some clarity to the proposed algorithm.

**Weaknesses:**

There is not a single equation in the paper, which I suppose is not a general requirement, but they are missed in this submission. I have provided examples below where equations would have been needed, but, as a general note on the quality of writing and structure of the paper, the absence of equations and of use of paragraphs are key components that make the submission difficult to follow. There are *large*, excessive blank spaces surrounding Fig. 1 and Alg. 1, which further down-weights the quality of the structure of the paper. As I describe below, I find the description of the BayesNJ approach (I am currently unsure if BayesNJ refers to a method) in Sec. 3.2-3.3 overly generic to the extent that I find Sec. 3.2 obsolete.


Regarding novelty, I think the construction of the distribution over merges is the same as the proposals used in the phylogenetic SMC algorithms (see e.g. Bouchard-Cotê et al. (2012), Moretti et al. (2022) or Koptagel et al. (2022)), and I am missing a comparison to the topology sampler in PhyloGFN (which also samples topologies by merging from leaves and up). As this is not the first likelihood-free phylogenetic inference algorithm (see e.g. Phyloformer) the novelty of Phyloformer 2 seems to lie in the EvoPF architecture. I think the submission should reformulate its contributions with this in mind.

My overall assessment is that this submission needs further work before it is ready for publication. The new EvoPF seems to be able to give promising inference time results, but the clarity of writing (structure of the text, use of paragraphs, inclusion of equations), a more refined communication of the methodological contributions and experiments on real data would greatly improve the submission. As such, I sincerely hope that the authors keep polishing their work and consider my (hopefully constructive) comments below!

Below follows in depth comments on the different section in the paper.

# Suggestions to improve the abstract
1. Why Phyloformer **2**? What is its predecessor?
2. "The first likelihood-free ... for posterior distributions".
	2. a) Then "a previous likelihood-free ... for point estimation". What does point estimation mean in this setting? The output of the algorithm is a single phylogeny?
	2. b) "Phyloformer 2 exploits a novel encoding for pairs of sequences that makes it more scalable than *previous approaches*". But Phyloformer 2 is the first method of the kind? What mechanism of the algorithm is the encoding part and how is it comparable to non-LF methods?
3. To me "Under realistic models of sequence of evolution" sends a conservative message, ironically suggesting that you do not use models that fit real data well but that are computationally realistic. I would rephrase it to emphasize that you are able to apply models that are more realistic *than the over-simplified models that are typically used in phylogenetic inference (like the JC model)*

# Introduction

In line 46-53 the text is unfortunately confusing, contains statements that are not motivated and that are incorrect (due to imprecision). My confusion is mainly induced by the intro lacking structure (it is largely one dense paragraph), while this following statement lacks motivation:

"Across all these approaches, a major hurdle is the computational cost of evaluating the likelihood function which is required for numerical optimization in maximum likelihood estimators, and to compute acceptance probability in sampling strategies or the evidence lower bound (ELBO) objective to be maximized in variational inference."

There needs to be a citation or a stand-alone explanation here. Furthermore, the sentence needs to be made more precise: I would assume that what is costly in the ELBO is the likelihood computation? The prior is typically uniform over the tree space and an exponential distribution over branch lengths, and generating trees with SLANTIS in VaiPhy or with SBNs in VBPI is a forward pass through a Bayesian net -- in both I believe you obtain the likelihood of sampling the tree from the proposal simultaneously with the tree generation? So, what remains is the likelihood function, not the other terms in the ELBO. Hence the statement is inaccurate. A similar argument holds for the acceptance probability calculation.

I am aware that the likelihood function is an obstacle for applying auto-differentation in phylogenetic inference through this paper https://arxiv.org/abs/2211.02168, so I suggest you read it and cite it here if you agree that it matches.

Due to the lack of paragraphs in the intro, the subsequent sentence (line 49) sort of suggests that likelihood-based methods and/or Bayesian methods explore all tree-topologies, which is definitely not true. It is then suggested that the phylogenetic community has made "this computation" -- the brute-force exploration of the phylogenetic tree space -- feasible, which I also do not think is correct? At least for any interesting size of $n$? If I am wrong, please provide a reference.

# Related work:
This is where Phyloformer (1) is first mentioned. Given that this submission proposes Phyloformer 2, I would have expected to see its predecessor mentioned earlier, and at this point of the submission I do still not see clearly why Phyloformer 2 is an appropriate name? This could be cleared out by repeating the mechanisms of Phyloformer 2 and distinguishing how it compares with its predecessor. E.g., is Phyloformer a likelihood free method? There is no mention of the use of a likelihood function when explaining PF.

Furthermore, "Given all correct pairwise distances, an existing algorithm (neighbor joining, NJ, Saitou & Nei, 1987) is guaranteed to reconstruct the correct tree, but the authors observed limited topological reconstruction accuracy[...]" in this sentence, "authors" is pointing to Saitou and Nei, which, in conjunction with the absence of paragraphs in the text overall, led me to believe that the text had moved away from Phyloformer and was now concerned with NJ as a related work. So, as in Introduction, there is a great need to divide the text into multiple paragraphs to improve legibility and structure.

Something I am missing in this section is where BayesNJ (or the procedure of constructing an approximation over phylogenies) differs from the merge-based approximations using in sequential Monte-Carlo-based phylogenetics and in PhyloGFN. For instance, how is distribution $q_m$ here different from the learnable merge proposal distribution in Moretti et al. (2022), or in \phi-SMC in Koptagel et al. (2022)? Is the difference mainly the different objective functions used to learn the parameters? A clear distinction to these approaches would also improve the exposition of BayesNJ.

# 2.1 Notation
There is an inconsistent use of $N$ and N to denote the number of leaves. Also, in the introduction $n$ denotes the number of taxa.

The text inside the hyphen, line 133-136, is difficult to parse.

"the parameters output by this neural network" --> outputted

In the KL divergence in line 152, the expectation is taken w.r.t. $p(x)$ which is not defined. Is this KL tractable? Probably not, so how is it "generally" minimized by simply minimizing the log-likelihood of the approximation on the samples generated by the model?

In line 159, what is the target average KL? If its the E[KL] term in 152, please give that term its own equation environment and reference it -- now target average KL is not defined.

I appreciate Figure 1 a lot. There is a lot of blank space above it though. I would remove it order to account for the additional space needed when updating the text with paragraphs.

# Section 3.
An overall comment on this section is that the explanations of the two new modules are overly generic to the extent that I cannot evaluate the feasibility of the methods. This is a pity since I was very excited to understand them. Below follow more detailed comments.

Minor: please spell out EvoPF before introducing the abbreviation.

Reading section 3.2, I have difficulties understanding exactly what BayesNJ refers to? Is it a distribution, an algorithm or something else? As far as I can see, $q_m$ and $q_\ell$ are not defined, so I am also struggling with understanding what distributions we are parameterizing here? In all I find this subsection obsolete.

Could you explain how the $\psi(x)$ is connected to the parameters of $q_m$ and $q_\ell$? Preferably already in Section 3.2, but also in Section 3.3, for example in line 268: the "softmin across pairwise scores" formulation could instead be an equation which shows how $\psi(x)$ (the scores?) is used to parameterize the distribution. I would recommend using an equation for it.

"We model the probability $q_s$ as a Beta distribution whose parameters are produced by a bilinear function of the embedding of the two merged nodes." This should again be an equation such that it is possible to grasp the relationship between $\phi(x)$ and the parameters of the Beta distribution. Also, what is the bilinear function?

The statement in line 284 regarding the factor that comes from the "determinant of the Jacobian of the change of variable" is not possible to understand without an equation, in my opinion. Could you write this out mathematically, please?

Lines 299-302 can be formulated using an equation instead which would make it easier to understand the greedy MAP approximation. Regarding the name "greedy MAP approximation", I am slightly confused: is this a greedy algorithm for approximating the MAP, is the MAP a greedy approximation or is the "greedy MAP" an approximation of something? In line 404 later, they are only referred to as greedy-MAP trees, not greedy MAP approximation trees.

# Section 4
"In order to fairly compare Phyloformer 2 (PF2) and the original Phyloformer (PF), we trained a version of Phyloformer 2 over the dataset used in Nesterenko et al. (2025) to evaluate PF". Use the introduced abbreviations consistently here.

I couldn't understand this: "≈ 170, 000 50-taxa tree/MSA pairs". Does it mean to say that there are 170,000 trees, each with $N=50$ taxa?

LG+GC is not defined or referenced. I could not find a definition of it in the Phyloformer paper either, and I have not heard of it before. Since all substitution models can be formulated mathematically, I would heavily advise to actually write it out so that the reader can evaluate the complexity of using this model instead of, e.g., the JC model which is heavily employed in the phylogenetic inference papers published in the machine learning community.

PF2_{\ell_1} is not defined.

"Kuhner-Felsenstein distances" are not defined in the text, and can arguably not be taken for granted as common knowledge -- how is it different from RF? In contrast the more standard RF distance is referenced.

Line 437: KF is not defined.

Line 449: LG+G8 is not defined (same comment as above regarding the mathematical formulation of the model.)


# Typos and necessary fixes

* The legibility of the introduction would greatly benefit from more structure, like using paragraphs (I mean here blank lines in tex, not \paragraph). For instance you could break up the dense chunk of text in the first paragraph with a newline at line 41.
* When mentioning VI-based approaches, I think VaiPhy needs credit as it was the first mean-field VI algorithm for phylogenies.
* line 55 imped -> impede
* line 68, a NPE -> an NPE
* "EvoFormer module used in Alphafold 2" needs citation.
*  byNesterenko et al. (2025) -->  by Nesterenko et al. (2025)
* $\ell_1$ is a notational clash with the branch length sets. You could use $L_1$?
* Typo line 424: "a slightly better topological accuracies"

**Questions:**

Given access to real data, how would you utilize it in order to get better posterior approximation?

---

> ### Author Response · Authors · 2025-11-21
> **Author response to reviewer 2.  (1/2)**
>
> We thank the reviewer for his particularly extensive review and the many constructive and helpful comments. We hope to address them in our response below.
>
> ## Weaknesses
>
>  - **(1)** We have improved the clarity of the methods, in particular by adding equations to sections 3.2 and 3.3. Overall we have added more paragraphs and linebreaks throughout the text to improve readability.
>  - **(2)** Our contribution goes beyond providing a new architecture for a problem already addressed in the previous literature: fundamentally, we propose the first method for *amortized* and *likelihood-free* model-based Bayesian inference of phylogenetic tree distributions. Phyloformer (Nesterenko et al.) is likelihood-free and amortized, but it does not estimate the posterior distribution of phylogenies, only the median of the posterior distribution of evolutionary distances. Our experiments show that targeting trees rather than distances makes Phyloformer 2 much more topologically accurate than Phyloformer.
> In addition, it provides a well-calibrated posterior distribution instead of a single tree *(see new Fig 3b)* . Previous variational phylogenetic inference approaches do estimate the posterior distribution of phylogenies, but they are ELBO-based and thus inherently committed to models with tractable likelihood. Similarly, previous non-variational SMC approaches are Bayesian but, through the use of importance sampling, require a tractable likelihood under the target model. Furthermore, both families of likelihood-based approaches are not amortized, making their high computational cost even more critical. We now clarify this very important point in the revision.
> We also mention SMC in 3.2 when we introduce or distribution (since the chosen criterion is indeed similar), and mention GFlowNets as a possible future extension. Indeed, this approach as implemented in PhyloGFN is committed to tractable models as it relies on conditional likelihoods to represent internal nodes so it could not be directly translated to our likelihood-free framework.
>  - **(Abstract)** We have clarified the abstract to address the reviewer's suggestions.
>  - **(Introduction)** We have broken up the introduction into several paragraphs, improving readability. We have also made it clear that the costly part both in maximum-likelihood and variational approaches using the ELBO is the computation of the likelihood.
> We have also updated the paper to clarify that we do not claim maximum-likelihood methods exhaustively explore the entire tree space but rather use heuristic search strategies *(see lines 47-56)*.
> Regarding the suggested reference to Fourment et al., we would like to stress that we present a likelihood-free method not because we cannot compute the gradient of the likelihood function but rather because we want to do phylogenetic inference in settings where we cannot compute the likelihood itself.
>  - **(Related Work)** We have added more information to this section (see response to weakness 2) and clarified how Phyloformer stands in this field. *(see lines 114-117)*
>  - **(Notation)** We have resolved inconsistencies in notation. We have also added motivation for using the average KL as a target. *(see lines 157-161)*
>  - **(Section 3)** We have extensively re-worked this section in order to clarify it. In 3.1 we have added a little more detail on the evoPF module and point to the supplementary material for a full characterization.
> In 3.2 we have explained what $q_\ell$ and $q_m$ are and mentioned that they are parametrized by $\psi(s)$ *(see lines 252-255)*, leaving details for 3.3 so that this section stays focused on how to define a proper distribution over phylogenetic trees.
> In 3.3 we have added equations formally defining $q_\ell$ (with components $q_s$ and $q_r$)  and $q_m$ and exactly how they are parametrized in the BayesNJ module.
> We have added section A.2 in the supplementary material discussing the change of variable and reference it in 3.3. We also added an equation to describe our "greedy MAP". *(see lines 325-331)*
> - **(Section 4)** We have added a better description of the training data with even more details in the supplementary material *(see A.3)*. We do not train on the 170k examples of PF but a large training set with $\approx1.3M$ trees each with 50 taxa. We have also added references to the LG model which is a standard amino acid substitution model, as JC is to nucleotide substitution models. We have also switched from GC site-heterogeneity for evolutionary rates to a discrete gamma with 8 categories (G8) and added the corresponding reference.
>
> We also thank the reviewer for catching small typos, we have corrected them in this revised manuscript.

---

> > ### Author Response · Authors · 2025-11-21
> > **Author response to reviewer 2. (2/2)**
> >
> > ## Questions
> >
> > Real data including alignments and their phylogeny could be used to define an empirical distribution of (alignment, tree). Phyloformer 2 could learn the corresponding posterior instead of the one corresponding to the probabilistic models currently used for simulation.
> > However, doing so would require a large dataset, and we believe it is more realistic to aim to doing phylogenetic inference under a probabilistic model (possibly a complex one thanks to our likelihood-free approach).
> > Given a moderately large empirical dataset, a possible intermediate strategy could also be to pre-train a network on data simulated under a probabilistic model, and fine-tune it on the empirical distribution.

---

### Official Review · Reviewer_3zMD · 2025-11-01

**Soundness:** 3
**Presentation:** 3
**Contribution:** 3
**Rating:** 4
**Confidence:** 4

**Summary:**

This paper introduces Phyloformer 2, a neural posterior estimation (NPE) method for phylogenetic inference. The approach combines two components: EvoPF, an encoder that transposes AlphaFold's EvoFormer architecture to process multiple sequence alignments at the sequence-pair level rather than position-pair level, and BayesNJ, a factorized posterior distribution over tree topologies and branch lengths. The authors demonstrate improved topological accuracy and computational speed compared to existing likelihood-based and likelihood-free methods on simulated datasets. The work represents an interesting application of modern deep learning to phylogenetic inference, though several fundamental limitations warrant careful consideration.

**Strengths:**

1. Clear methodological contribution: The factorization of the posterior over successive merges (BayesNJ) is conceptually elegant and provides a tractable way to define distributions over tree space, addressing a genuine challenge in neural posterior estimation for discrete structures.

2. Comprehensive experimental evaluation: The paper includes comparisons with multiple established methods (IQTree, FastTree, FastME, Phyloformer) across various metrics and provides ablation studies distinguishing the contributions of EvoPF and BayesNJ.

**Weaknesses:**

1. Computational cost accounting is incomplete: While the paper emphasizes inference speed advantages, the amortization trade-off deserves fuller discussion. Training requires generating ~170,000 simulated tree/MSA pairs plus additional fine-tuning data. Although forward simulation is cheaper than likelihood evaluation, the total computational budget (simulation + GPU training time) should be reported. The paper should clarify: (a) total training wall-clock time, (b) the break-even point where training cost is amortized over multiple inference tasks, and (c) computational requirements when adapting to new priors or models.

2. Strong distributional assumptions not prominently disclosed: The method requires training data from a specific prior and evolutionary model. While this is mentioned (lines 83-89, 485-489), phylogenetics researchers unfamiliar with NPE may not immediately recognize this as a fundamental constraint rather than a technical detail. A brief statement in the abstract or introduction highlighting that the method is 'model-specific and requires retraining for different evolutionary scenarios' would help set appropriate expectations. This trade-off—fast amortized inference at the cost of model commitment—should be positioned as an inherent characteristic of the NPE paradigm rather than a limitation unique to this work.

3. Scalability claims require context: The paper demonstrates clear scalability improvements over Phyloformer: memory usage is reduced by approximately 50% (Figure 2c), and the method can handle 300 sequences of length 250—a regime inaccessible to the original Phyloformer. However, contextualization against likelihood-based methods would strengthen this claim. How do these limits compare to typical problem sizes in modern phylogenetics? For instance, IQTree can handle thousands of sequences, though at much higher computational cost. Clarifying whether PF2's current scalability addresses common use cases or remains limited to smaller-scale problems would help readers assess practical applicability.

4. Limited validation of posterior quality: Figure S.2 shows differences between PF2 and RevBayes posterior samples, with PF2 producing more diffuse bipartition probabilities. The authors suggest RevBayes might be miscalibrated, but provide no evidence for this claim. Proper calibration assessment (e.g., using simulation-based calibration) is needed to validate the posterior approximation quality.

5. Novelty of EvoPF module is overstated: The module is described as "novel" but is explicitly acknowledged as "inspired by" (i.e., adapted from) AlphaFold2's EvoFormer with "a few simplifications." The contribution appears to be primarily an architectural adaptation rather than a fundamental innovation. This should be characterized more accurately.

**Questions:**

* Training cost transparency and amortization analysis: Can you provide (a) total wall-clock time and computational resources for training across all experiments, (b) a break-even analysis showing how many inferences are needed to recover training costs, and (c) the cost of adapting to new evolutionary models? For a research project analyzing 100 datasets under the same model, how does the total computational budget compare to using likelihood-based methods?

* Out-of-distribution performance quantification: Have you systematically evaluated performance when test data comes from evolutionary models not seen during training, or when the true prior differs from the training prior? The current treatment of this critical limitation is too brief. Quantitative results showing degradation patterns would help users understand when the method should or shouldn't be applied.

* Canonical ordering justification and alternatives: The canonical merge ordering (Section 3.2) is presented as necessary to avoid summing over all possible orderings. However, couldn't a permutation-equivariant architecture potentially avoid this constraint while maintaining tractability? Have you explored whether relaxing this ordering requirement might improve posterior expressivity, even if it increases computational cost?

* Posterior calibration methodology: Beyond the single-alignment comparison with RevBayes (Figure S.2), have you performed systematic calibration checks? For example, simulation-based calibration where you: (1) sample parameters from the prior, (2) simulate data, (3) obtain posterior samples from PF2, (4) verify that true parameters are uniformly distributed in posterior quantiles? This would help determine whether the diffuse posteriors reflect uncertainty or miscalibration.

---

> ### Author Response · Authors · 2025-11-21
> **Author response to reviewer 1.**
>
> We thank the reviewer for their thorough comments, please find our response below.
>
> ## Weaknesses
>
> - **(1)** This is a valid concern, since the experiments require re-training a version of PF2 we will address this in our final revision.
> - **(2)** We have added a sentence stressing this point in the conclusion of the paper *(see lines 481-488)*. It is important to note however that this phenomenon is not limited just to NPE. Indeed, Maximum-likelihood and Bayesian methods are also model-based and rely on strong assumptions about the evolutionary processes.
> A method for addressing this problem is testing the models for goodness-of-fit (e.g. with IQTree's ModelFinder) but we believe it is outside the scope of this work.
> - **(3)** While scalability is indeed an issue, we want to stress that we do not claim that Phyloformer 2 is scalable to modern phylogenetic inference sizes. Rather, what we present here is a first conceptual basis for amortized model-based phylogenetic NPE.
> While there is assuredly more work to do to make it scalable, we believe that being more accurate and faster than Phyloformer and IQTree for a moderately big number of taxa suggests that the approach is promising for ultimately doing phylogenetic inference under intractable models for larger problem sizes.
> - **(4)** We agree with the reviewer on that point. To address this we have added a simulation-based calibration study to the results sections as well as a supplementary figure. *(see Fig 3b and lines 463-468)*
> - **(5)** It is of note that we are, to our knowledge, the first to introduce this architecture for phylogenetic NPE, and Figure S.2 indicates that it an instrumental part of Phyloformer 2's performance. Therefore we believe that this architecture is indeed novel for phylogenetic inference.
>
> ## Questions
>
> - **(1)** Please see our response to weakness 1.
> - **(2)** Please see our response to weakness 2, and similarly our response to Weakness 1 of reviewer 3.
> - **(3)** In order to be equivariant to permutations of merges, each merge would need to be independent of all other merges. We believe that this would remove a lot of information useful to tree reconstruction. We have tried experiments without enforcing the order at inference time that only marginally impacts topological accuracy. However, enforcing the order does not significantly increase computational time and leads to a theoretically sound and well-defined distribution.
> In the current literature on (non-amortized) variational phylogenetic inference or SMC, this problem is addressed in different ways: (1) by imposing constraints similar to ours [1], in a rooted or unrooted context; (2) by relying on a pre-defined order [2], but then this entails an entirely different intermediate representation that requires to consider, not just the tips still to be merged, but also the tree constructed thus far; (3) using GFlowNet [3], which provides an elegant framework for implicitly dealing with the intractable sum over all possible merges, but is limited in its current form to tractable likelihoods; (4) by ignoring the problem altogether [4,5], thus not unlike our attempt to just relax the constraint, but again, at the cost of not targeting the originally intended posterior.
> - **(4)** Please see our answer to weakness 4.
>
>
> ### References:
> [1] Bouchard-Côté et al., Phylogenetic inference via sequential monte carlo, 2012 (https://doi.org/10.1093/sysbio/syr131).
> [2] Xie and Zhang, ARTree: A deep autoregressive model for phylogenetic inference, 2023 (https://openreview.net/forum?id=SoLebIqHgZ)
> [3] Zhou et al., PhyloGFN: Phylogenetic inference with generative flow networks, 2024 (https://openreview.net/forum?id=hB7SlfEmze)
> [4] Koptagel et al., Vaiphy: a variational inference based algorithm for phylogeny, 2022 (https://proceedings.neurips.cc/paper_files/paper/2022/file/5e956fef0946dc1e39760f94b78045fe-Paper-Conference.pdf)
> [5] Moretti et al., Variational Combinatorial Sequential Monte Carlo Methods for Bayesian Phylogenetic Inference, 2021  (https://proceedings.mlr.press/v161/moretti21a/moretti21a.pdf)

---

> > ### Comment · Reviewer_3zMD · 2025-11-24
> >
> > Dear Authors,
> >
> > Thank you for your response. I note that the rebuttal phase typically allows paper updates. Could you please clarify which revisions have already been implemented in the current manuscript (e.g., the simulation-based calibration study, conclusion modifications) versus which are planned for the camera-ready version (e.g., computational cost experiments)? Please provide specific section/figure numbers for completed changes so I can re-evaluate them and potentially adjust my assessment accordingly.
> >
> > Best regards,

---

> > > ### Author Response · Authors · 2025-11-25
> > >
> > > Dear reviewer,
> > >
> > > we had, it seems erroneously, thought that you would be provided with a diff highlighting the differences between the initial and our revised manuscript.
> > > We have updated our responses to all reviewers to include line, section and figure numbers in the revised manuscript, indicating where we have significantly changed the text.
> > > We have also just uploaded a new version of the revised manuscript, as the first revision did not contain one of the changes to the conclusion, which we mention in our response to your comments.
> > >
> > > We hope that you find these additional details helpful in re-evaluating our submission.
> > >
> > > Best regards,

---

> ### Comment · Reviewer_3zMD · 2025-11-26
>
> Dear Authors,
>
> Thank you for the detailed response and the updated manuscript with clear references to all changes.
>
> I appreciate your efforts in addressing the reviewers' feedback. While not all concerns have been completely resolved, the substantial improvements made have significantly enhanced the manuscript quality. Accordingly, I have raised my score.
>
> Best regards,
> Reviewer 3zMD

---

### Author Response · Authors · 2025-11-21
**Global Author response**

We thank the reviewers for their feedback and constructive criticism. We provide below a point by point answer to the reviewer's comments addressing improvements in the clarity and overall writing quality of the paper. In particular we:

  1. Added more context on deep-learning in phylogenetic inference and Bayesian phylogenetics, highlighting how Phyloformer 2 is situated in this landscape and its novel aspects
  1.  Clarified the methods section, adding equations and details
  1.  Added a small calibration study to the results section

In this revision, we mainly addressed concerns about the text of the paper itself. We are currently running additional experiments (mainly aimed at testing Phyloformer 2 on empirical data). We aim to present a new revision of this paper including these results as soon as we can, leaving time for the reviewers to take that into account before the final deadline.

---

### Author Response · Authors · 2025-12-04
**Final author response (1/2)**

We thank the reviewers again for their valuable feedback and constructive criticism. We are of course disappointed that this discussion phase has been impacted by security issues, however we believe that we have sufficiently addressed their concerns in this final version of the article, and that they would have been satisfied with our added work.

In order to make the Area chair's work easier, we list below the main changes and additions to the paper and their effect on the quality of the paper. More detailed lists of changes are also listed in our responses to reviewers.

## Clarity and writing related changes

In answer to some criticism from several reviewers, we have re-worked the introduction and related-work sections. We believe these changes make the current context of deep-learning for phylogenetic inference and Bayesian phylogenetics clearer. Furthermore, these changes also highlight how Phyloformer 2 is situated in this landscape and its novel aspects.

In response to reviewer Jzst's extensive (and very valuable) review, we have extensively reworked the sections presenting the various methodological aspects of Phyloformer 2. In particular we have cleared up and harmonized the notation, broken up text into smaller thematically consistent paragraphs, added clarification as to how the model parametrizes the probability distributions and have made more extensive use of equations.
Overall we believe that the suggestions from reviewer Jzst have greatly improved the quality of the paper and its clarity.

In response to several reviewer remarks, we have also added more information on the training and testing datasets in the results section as well as the supplementary material.

Finally ---to address reviewer concerns--- we have clarified the conclusion to indicate how Phyloformer 2 behaves in scenarios where testing data is significantly different from training data, and more importantly how it compares to other methods in that respect.

## New experiments and results

To address the remarks of reviewer 3zMD we have:
  - Added a break-even analysis section in the supplementary material and mention it in the results section. We also indicate training and fine-tuning times. We conclude that the break-even point for point estimation with maximum likelihood methods is quite high because of the high upfront cost of simulation. However, this is significantly lower when sampling large numbers of trees from the posterior.
  - Conducted a simulation-based calibration study that we have added in the main text (figure 3b). This shows that the learned posterior distribution that PF2 samples from is well calibrated.

To address concerns from reviewer 7J8h, we have also examined how sensitive PF2 estimates are to alignment length. We have added the results in supplementary figure S.6. These new results show that PF2 is much less sensitive to alignment length than to the number of sequences in the alignment.

## Real data experiments

All three reviewers have mentioned the need for experiments on empirical biological data.
We simulated new large training and fine-tuning datasets with gap proportions more closely matching one of the empirical dataset used in the Phyloformer (PF1) paper (borrowed from Zhou *et al.*, 2018, [10.1093/molbev/msx302](http://doi.org/10.1093/molbev/msx302)). However, after several attempts at fine-tuning models under different tree priors we have not managed to get satisfactory results on this dataset. On the contrary, we obtained results matching those of PF1 on the other empirical dataset (borrowed from Szöllosi *et al.* 2013 [10.5061/dryad.pv6df](http://doi.org/10.5061/dryad.pv6df)).

As of right now, we do not know why PF2 performs worse than PF1 on the Zhou *et al.* dataset. We are confident that given enough time we should be able to understand the cause better and would rather wait to publish a discussion of these experiments that would be useful to the readers.

We would like to stress that Phyloformer 2 should not necessarily be viewed as a ready to use tool, but rather as a methodological proof of concept and stepping stone. Although it may not be immediately useful, we believe that this work could lay the foundation to more efficient methods that could open the door to fast phylogenetic inference and posterior sampling under currently inaccessible evolutionary models.
It also seems to us that outperforming likelihood-based methods under the LG model (whose likelihood can be computed) and showing that our performance gain further increases under Cherry (whose likelihood cannot be computed efficiently) and providing a well-calibrated posterior distribution is already a strong result validating the interest of Phyloformer 2.

---

> ### Author Response · Authors · 2025-12-04
> **Final author response (2/2)**
>
> ## Final remarks
>
> We have substantially changed and reworked the paper, adding experiments and clarifying many sections. We believe that this version of the manuscript is of much higher quality than the initial submission in part because of the in depth and pertinent reviews we have received.
> We hope that ---even though the reviewers cannot comment on these changes--- they are satisfied with the changes and would agree with us that this work has a place at this venue.
> We also thank the area chair for their work and hope this new version of our article is up to their standards.
>
>
>
> *PS: We would also like to note that we have a repository containing the code for this paper, but have not included it in the article in order to keep anonymity intact.*

---

### Meta-Review · Area_Chair_rtJa · 2026-01-07

**Summary:**

This paper introduces Phyloformer 2, a likelihood-free, amortized neural posterior estimation method for phylogenetic inference that directly approximates posterior distributions over tree topologies and branch lengths. The approach combines an EvoFormer-inspired encoder for multiple sequence alignments (EvoPF) with a merge-based factorization of tree distributions (BayesNJ). On simulated datasets, the method demonstrates improved topological accuracy and substantial inference-time speedups compared to both likelihood-based methods and prior neural approaches, while also providing posterior uncertainty estimates.

**Reviewer Concerns:**

Several reviewer concerns were largely addressed in the rebuttal. The authors substantially improved clarity and presentation by adding missing equations, clearer definitions, and better organization of the methods sections. The addition of simulation-based calibration strengthens the validation of posterior quality and addresses earlier concerns about diffuse uncertainty estimates. The rebuttal also clarifies the relationship to prior work (SMC-, VI-, and GFlowNet-based approaches) and more carefully frames the novelty claims.

However, important limitations remain. Scalability is still restricted, with performance degrading beyond ~100 taxa and a hard implementation limit around ~200 taxa. The paper does not provide a transparent accounting of training and amortization costs, making it difficult to assess practical trade-offs. Robustness to model or prior mismatch and performance on real data remain largely unexplored. While the canonical merge ordering is better justified, it still constrains posterior expressivity and is deferred to future work.

**Reviewer Scores:**

Reviewer 3zMD is likely to raise the score as promised. The others are likely to keep their scores.

---

### Decision · Program_Chairs · 2026-01-26

Reject